# Gene identification and RNAi-silencing of p62/SQSTM1 in the vector *Rhodnius prolixus* reveals a high degree of sequence conservation but no apparent deficiency-related phenotypes in vitellogenic females

Jéssica Pereira[1], Samara Santos-Araujo[1], Larissa Bomfim[1], Katia Calp Gondim[1], David Majerowicz[2,3], Attilio Pane[4], Isabela Ramos[1] *

1 Instituto de Bioquímica Médica Leopoldo de Meis, Universidade Federal do Rio de Janeiro, Rio de Janeiro, RJ, Brazil, 2 Departamento de Biotecnologia Farmacêutica, Faculdade de Farmácia, Universidade Federal do Rio de Janeiro, Rio de Janeiro, RJ, Brazil, 3 Programa de Pós-Graduação em Biociências, Universidade do Estado do Rio de Janeiro, Universidade Federal do Rio de Janeiro, Rio de Janeiro, RJ, Brazil, 4 Instituto de Ciências Biomédicas, Universidade Federal do Rio de Janeiro, Rio de Janeiro, RJ, Brazil

* isabela@bioqmed.ufrj.br

## Abstract

Autophagy and the ubiquitin-proteasome system (UPS) are important cellular mechanisms that coordinate protein degradation essential for proteostasis. P62/SQSTM1 is a receptor cargo protein able to deliver ubiquitinated targets to the proteasome proteolytic complex and/or to the autophagosome. In the insect vector of Chagas disease, *Rhodnius prolixus*, previous works have shown that the knockdown of different autophagy-related genes (ATGs) and ubiquitin-conjugating enzymes resulted in abnormal oogenesis phenotypes and embryo lethality. Here, we investigate the role of the autophagy/UPS adaptor protein p62 during the oogenesis and reproduction of this vector. We found that *R. prolixus* presents one isoform of p62 encoded by a non-annotated gene. The predicted protein presents the domain architecture anticipated for p62: PB1 (N-term), ZZ-finger, and UBA (C-term) domains, and phylogenetic analysis showed that this pattern is highly conserved within insects. Using parental RNAi, we found that although p62 is expressed in the ovary, midgut, and fat body of adult females, systemic silencing of this gene did not result in any apparent phenotypes under in-house conditions. The insects' overall levels of blood meal digestion, lifespan, yolk protein production, oviposition, and embryo viability were not altered when compared to controls. Because it is known that autophagy and UPS can undergo compensatory mechanisms, we asked whether the silencing of p62 was triggering adaptive changes in the expression of genes of the autophagy, UPS, and the unfolded protein response (UPR) and found that only ATG1 was slightly up regulated in the ovaries of silenced females. In addition, experiments to further investigate the role of p62 in insects previously silenced for the E1-conjugating enzyme (a condition known to trigger the upregulation of p62), also did not result in any apparent phenotypes in vitellogenic females.

**Data Availability Statement:** All relevant data are within the manuscript and its Supporting Information files

**Funding:** "This research was funded by the following grants: Jovem Cientista do Nosso Estado (JCNE)- Fundação de Amparo à Pesquisa do Estado do Rio de Janeiro (FAPERJ) (www.faperj. br/), Instituto Nacional de Ciência e Tecnologia em Entomologia Molecular (INCT-EM)-Conselho Nacional de Desenvolvimento Científico e Tecnológico (CNPq) (http://cnpq.br/) and Coordenação de Aperfeiçoamento de Pessoal de Nível Superior (CAPES) (www.capes.gov.br/) to I. R. The funders had no role in study design, data collection and analysis, decision to publish, or preparation of the manuscript."

**Competing interests:** The authors have declared that no competing interests exist.

## Introduction

*Rhodnius prolixus* is a strictly hematophagous hemipteran in which a number of basics aspects of insect physiology were discovered early in the last century as well as its health importance as a vector of the etiological agent of Chagas Disease, a neglected tropical disease endemic in Central and South America [1, 2]. Currently, Chagas disease affects more than 8 million people and estimates are that climate change and globalization are likely to further expand its endemic area [3, 4].

In eukaryotic cells, the fine-tuning of translation or degradation pathways controls proteostasis, and its disturbance typically leads to cellular dysfunction and disease. Multiple studies have shown that autophagy and the ubiquitin-proteasome system (UPS) are the most important cellular post-transcriptional mechanisms that coordinate protein degradation, playing an essential role in general cellular proteostasis [5]. The UPS conjugates the protein ubiquitin to specific proteins as a signal that routes the ubiquitinated targets for degradation through the proteasome proteolytic complex. The autophagy pathway generates double-membrane vesicles named autophagosomes that engulf targets such as protein aggregates and whole damaged organelles, delivering them for degradation in the lysosome [6, 7].

Due to their highly different types of machinery and degradation sites, it has been established that, canonically, the UPS favors targeting individual proteins, mainly short-lived, misfolded, and damaged proteins, whereas autophagy has the capacity to degrade larger targets such as macromolecule complexes and entire organelles [8]. However, in the past years, it has been repeatedly observed across different models that autophagy is enhanced as a compensatory mechanism for deficient UPS and vice versa, thus revealing a crosstalk between both pathways to ensure efficient proteostasis [9]. In addition to the compensatory mechanisms, it has been observed that those pathways also crosstalk at the machinery level, where cargo receptors for selective autophagy can deliver ubiquitinated targets to nucleating autophagosomes, for eventual degradation via autophagy. Those receptor proteins usually share a conserved domain architecture including a LIR motif, which mediates its interaction with LC3/ATG8 family proteins in the autophagosomes, and a C-terminal UBA domain (ubiquitin interacting domain), which mediates its interaction to ubiquitinated targets [10].

Sequestosome-1 (SQSTM1 or p62), hereafter referred to as p62, is one of the most studied cargo receptors for selective autophagy of ubiquitinated targets. p62 has multiple conserved domains that are able to interact with various proteins with diverse functions. p62 conserved architecture includes the PB1 domain, important for oligomerization, the ZZ domain, that recognizes degrons in autophagic substrates, the LC3/ATG8- and Keap1-interacting motifs (LIR and KIR, respectively) and the C-terminal ubiquitin-binding UBA domain [10–12]. Thus, p62 is a ubiquitous multifunctional protein, which can direct ubiquitinated proteins to the proteasome [13, 14] or to the maturing autophagosome [15], and the regulations that manage the delivery to each of the pathways still remain enigmatic.

In humans the gene encoding p62 is systemically expressed and its malfunction is associated with a wide range of diseases, including neurodegenerative and metabolic diseases and multiple forms of cancer [16]. In insects, p62 sequence and function have been investigated in *Drosophila*, where one single isoform of a p62 homologue was found and named Ref(2)P (refractory to Sigma P ref(2)P/CG10360). It has 599 amino acids and contains an N-terminal PB1 domain followed by a ZZ-type zinc finger domain and a C-terminal UBA domain, as well as a predicted LIR motif between amino acids 451–558 (DPEWQLID) [17]. In functional analyses, inhibition of Ref(2)P in *Drosophila* lead to increased lifespan and declined motor

function [18], whereas up-regulation of Ref(2)P in mid-life delays the onset of pathology and prolongs healthy lifespan [19].

In *R. prolixus*, the RNAi-mediated maternal silencing of several genes of the autophagy and the UPS machineries have resulted in varied but recurring oogenesis-related phenotypes, generating drastically decreased oviposition rates and embryo lethality. Deficiencies in the autophagy-related genes (ATGs) ATG1/ULK1 and ATG3 resulted in chorion (eggshell) biogenesis abnormalities [20, 21], whereas the silencing of ATG6/Beclin1 generated impaired yolk endocytosis [22, 23]. Regarding the UPS, the silencing of polyubiquitin was maternally lethal, while the knockdown of highly abundant isoforms of the ubiquitin-conjugation enzymes E1 and E2 resulted in drastic chorion-related oogenesis abnormalities, as well as the upregulation of different ATGs [24]. Interestingly, it was found that the silencing of the ubiquitin activating enzyme E1 also resulted in the upregulation of p62 expression, presumably as a compensatory mechanism to adapt to the E1-knockdown effects [24].

Because the UPS and autophagy are essential intracellular degradation routes and p62 is a key mediator that allows direct crosstalk between these pathways, in this work we investigate the silencing effects of p62 in the context of *R. prolixus* oogenesis. Bioinformatics showed that the genome of *R. prolixus* encodes one single isoform of a p62 orthologue, presenting the typical p62 conserved domains: PB1, ZZ and UBA. In addition, we found that although p62 is systemically expressed in the different organs of the adult vitellogenic females, maternal RNAi silencing of this gene resulted in no apparent phenotypes in the insect digestion, lifespan, oocyte maturation or embryo viability. To further explore the role of p62, we tested the silencing of this gene in a condition where we know it is upregulated, in E1-silenced insects [24]. Even in the setting of E1 silenced insects, the knockdown of p62 did not result in substantial additional phenotypes, indicating that it is not essential for the proteostasis adaptation mechanisms triggered by the silencing of E1. Altogether, we found that p62 is conserved in *R. prolixus*, but it is not essential for the adult insect general physiology under in house conditions.

## Materials and methods

### Insects

Insects were maintained at a 28 ± 2˚C controlled temperature and relative humidity of 70–80%.

The insects are fed for the first time as adults in live-rabbit blood 14 to 21 days after the 5th instar nymph to adult ecdysis. After the first blood feeding, the insects (male and females) are kept back together to mate and generate the eggs that will hatch as first instar nymphs to maintain the insectarium. We know that the females were mated because we do monitor their oviposition rates and F1 eclosion rates during this cycle (virgin females do not lay many eggs, and they do not hatch because they are not fertilized). After this first blood feeding, all adult females are fed every 21 days and only fully gorged insects (allowed to feed at free demand, usually gaining 6–7 times the insect's initial body weight in 20–30 min) are used for the experiments. Thus, females of the second or third blood feeding were used, and they are mated, and highly synchronized regarding blood feeding, digestion, and oviposition. All animal care and experimental protocols were approved by the guidelines described in the ethics statement.

### Ethics statement

All animal care and experimental protocols were approved by guidelines of the institutional care and use committee (Committee for Evaluation of Animal Use for Research from the Federal University of Rio de Janeiro, CEUA-UFRJ #01200.001568/2013-87, order number 155/13), under the regulation of the national council of animal experimentation control (CONCEA).

## Gene identification

The sequence of *Rhodnius prolixus* p62/SQSTM1 was not annotated in the most up to date version of the *R. prolixus* genome available at Vector Base ([www.vectorbase.org](www.vectorbase.org)). The gene is hosted in the contig KQ034371 and was identified by similarity of the predicted transcript to the *Drosophila melanogaster* p62 ortholog sequence (Ref(2)P/ CG10360) using tBlastn. Schematic diagrams were generated using the DOG 2.0 software [25].

## Phylogenetic analysis

The genomes of *R. prolixus* [26], *D. melanogaster* [27], the mosquitoes *Anopheles gambiae* [28] and *Aedes aegypti* [29], the bee *Apis mellifera* [30], the postman butterfly *Heliconius melpomene* [31], the beetle *Tribolium castaneum* [32], the aphid *Acyrthosiphon pisum* [33], the whitefly *Bemisia tabaci* [34], the bed bug *Cimex lectularius* [35], the termite *Zootermopsis nevadensis* [36], and the water flea *Daphnia pulex* [37] were explored. All proteins containing the Pfam domain [38] PF00564 (PB1 domain) were obtained from the EnsemblMetazoa database [39] using the BioMart tool [40]. *R. prolixus p62/SQSTM1* gene was obtained as described above. The primary sequences were aligned with the Clustal W algorithm [41], and the phylogenetic analysis was performed by the maximum likelihood method [42] with 500 bootstrap repetitions in the MEGA 11 software [43]. *D. pulex* sequences were included as external group.

## *R. prolixus* p62 protein model

The predicted *R. prolixus* p62 protein 3D model was predicted using he software I-TASSER [44–46].

## Extraction of RNA and cDNA synthesis

All samples were homogenized in Trizol reagent (Invitrogen) for total RNA extraction. Reverse transcription reaction was carried out using the High-Capacity cDNA Reverse Transcription Kit (Applied Biosystems) using 1˚g of total RNA (after RNase-free DNase I (Invitrogen) treatment and checking RNA integrity in 2% agarose gels), Multiscribe Reverse Transcriptase enzyme (2.5 U/˚L) and random primers for 10 min at 25˚C followed by 2 hours of incubation at 37˚C. As a control for the DNAse treatment efficiency, we performed control reactions without the enzyme followed by testing the capacity of amplification by PCR. All samples were dissected 7 days after the blood meal.

## PCR/RT-qPCR

Specific primers for *R. prolixus* p62 were designed (S1 Table) and used in a PCR with the following cycling parameters: 10 min at 95˚C, followed by 35 cycles of 30 s at 95˚C, 30 s at 52˚C and 30 s at 72˚C, and a final extension of 15 min at 72˚C. Quantitative real-time PCR (RT-qPCR) was performed in a StepOne Real-Time PCR System (Applied Biosystems) using SYBR Green PCR Master Mix (Applied Biosystems) under the following conditions: 10 min at 95˚C, followed by 40 cycles of 15 s at 95˚C and 45 s at 60˚C. The relative expressions were calculated using the delta $C_t$ (cycle threshold) obtained using the reference gene 18S (RPRC017412) [47] and calculated $2^{-dCt}$ [48]. According to the minimum information for publication of quantitative RT-qPCR experiments (MIQE) Guidelines, normalization against a single reference gene is acceptable when the investigators present clear evidence that confirms its invariant expression under the experimental conditions [49]. The invariant expression of the reference gene 18S in this model under our conditions has been previously reported in [20, 23, 24, 50]. All other primers were validated before in the following papers [20–22, 24].

## RNAi silencing

dsRNAs were synthesized by MEGAScript RNAi Kit (Ambion Inc) using primers for p62 specific gene amplification with the T7 promoter sequence (S1 Table) targeting a fragment of 606 bp. Adult females were injected between the second and third thoracic segments using a 10˚l Hamilton syringe with 1˚g dsRNA (diluted in 1˚l of water) 2 days before the blood feeding. All phenotypes were observed over the following gonotrophic cycle (S1 Fig). A fragment of 808 bp of the *E. coli MalE* gene (Gene ID: 948538) included in the control plasmid LITMUS 28iMal obtained from the HiScribe RNAi Transcription kit (New England BioLabs) was amplified by PCR using a T7 promoter-specific primer, targeting the opposing T7 promoters of the vector. The cycling conditions were: 10 min at 95˚C, followed by 35 cycles of 30 s at 95˚C, 30 s at 52˚C and 60 s at 72˚C and a final extension of 15 min at 72˚C. The amplified fragment was used as a template for the synthesis of the control dsRNA (dsMal) [51]. All adult females injected with dsRNA were fed and transferred to individual vials. The mortality rates and the number of eggs laid by each individual were recorded daily and weekly, respectively.

## Hemolymph extraction and SDS-PAGE

The hemolymph was extracted from control and silenced females, 7 days after the blood meal, as originally described by Masuda e Oliveira, 1985 [52]. Approximately 10˚l of hemolymph per female was obtained by cutting one of the insects' legs and applying gentle pressure to the abdomen. The hemolymph was collected using a 10˚l pipette plastic tip. Once collected, the hemolymph was diluted 2x in phosphate buffered saline (PBS) 137 mM NaCl, 2.7 mM KCl, 10 mM $Na_2HPO_4$, and 1.8 mM $KH_2PO_4$, pH 7.4 and approximately 8 mg of phenylthiourea. The equivalent of 1˚l of hemolymph was loaded in each lane of a 10% SDS-PAGE.

## Determination of protein content

The total amount of protein in the midgut and hemolymph samples was measured by the Lowry (Folin) method, using as standard control 1–5˚g of bovine serum albumin (BSA) [53] in a E-MAX PLUS microplate reader (Molecular devices) using SoftMax Pro 7.0 as software.

## Yolk organelles suspension and flow cytometry

Flow cytometry was performed as previously described by [23]. Briefly, suspensions of yolk organelles were obtained by gently disrupting of recently dissected chorionated oocytes in PBS (2 oocytes in 250˚l of PBS). The population profiles of the yolk organelles were acquired on a FACS Calibur instrument (BD Bioscience, USA) powered by CellQuest Pro software v5.1 and analyzed using Flowing Software 2.5.1.

## Scanning electron microscopy

0–24 h laid eggs were carefully collected and fixed by immersion in 2.5% glutaraldehyde (Grade I) and 4% freshly prepared formaldehyde in 0.5 M cacodylate buffer, pH 7.3. Samples were washed in cacodylate buffer, dehydrated in an ethanol series, and coated with a thin layer of gold as previously described [54]. Models were observed in a Zeiss EVO 10 scanning electron microscope operating at 10 kV.

## Production of anti-*R. prolixus* p62 antibodies

Specific polyclonal antibodies for the single isoform of *R. prolixus* p62 were raised commercially by GeneScript. Rats were immunized with a 14-amino acid peptide ($NH_2$-WRDREGDQ

VQIWSC-COOH) derived from the predicted p62 sequence. The anti-p62 antiserum with an ELISA title of 1:512.000 was used for immunoblottings.

## Immunoblotting

Control (dsMal), as well as insects silenced for the autophagy related gene 6 (ATG6) [22], the autophagy related gene 8 (ATG8) [50] and p62 were obtained, and ovaries were dissected and homogenized using a glass/Teflon potter Elvehjem homogenizer in PBS. The homogenates, containing 60 ˚g of total protein, were separated by a 10% SDS-PAGE, transferred to nitrocellulose membranes, and blotted using antibodies against *R. prolixus* p62. Membranes were blocked in TBST (Tris 50 mM, pH 7.2, NaCl 150 mM, 0.1% Tween 20) containing 5% dry skimmed milk for 12 h. Primary antibodies were diluted 1:2500 (Anti-p62, described above) in the same buffer and incubated with the membranes for 2h. The membranes were washed 3x for 5 min and then incubated with the secondary antibodies (Goat Anti-Rat IgG H&L HRP, AbCam #ab7097) diluted 1:2000 for 2 h. After washing, the membranes were developed using the Pierce™ ECL Western Blotting Substrate.

## Results and discussion

### The single isoform of *R. prolixus* p62 is conserved within insects and is expressed in the ovaries, developing oocytes, midgut and fat body of adult vitellogenic females

The nucleotide sequence of *R. prolixus* p62 was identified as a non-annotated gene located in the contig KQ034371 listed in the present *R. prolixus* genome assembly version (RpC3.3) available at Vector Base (www.vectorbase.org). The full 14.5 kb sequence includes 8 exons and 7 introns generating a predict transcript of 1488 bp (S2 Fig). Phylogenetic analyses confirmed the identity of the *R. prolixus* gene as p62 (Fig 1, asterisk, red clade), as this sequence grouped in a clade with the *D. melanogaster* Ref(2)P/p62 gene (FBgn0003231) (Fig 1, red clade). Interestingly, this gene is duplicated in the genomes of *A. gambiae* and *Z. nevadensis* and triplicated in *D. pulex*. These duplication events are recent, considering the position and size of the genes in the clade and the tandem arrangement in the *A. gambiae* genome. On the other hand, the gene is apparently absent in *A. aegypti*, *H. melpomene*, *T. castaneum*, and *A. pisum*. Whether the gene was lost in the evolutionary process or was just not annotated in the genomes needs to be further investigated (Fig 1, red clade). Phylogenetic analyses also allowed the identification of other proteins with the PB1 domain. The RPRC013230 gene is orthologous to the *D. melanogaster* par-6 gene, involved in processes of cell polarization and tracheal branching [55, 56]. This gene is present in all analyzed genomes but duplicated in *B. tabaci* (Fig 1, blue clade). In turn, the RPRC000881 gene is an ortholog of the *D. melanogaster aPKC* gene, involved in neuronal proliferation [57]. This gene has orthologs in all analyzed genomes except *A. pisum* (Fig 1, green clade). Finally, RPRC008371 also has a PB1 domain and forms a clade with genes from *D. pulex*, *T. castaneum*, *A. pisum*, *B. tabaci*, *Z. nevadensis*, and *C. lectularius*. However, its function is still unknown (Fig 1, black clade).

The predicted transcript encodes a 495 amino acid protein presenting the anticipated conserved domain architecture of p62, including an N-terminal PB1 (IPR000270), a Znf_ZZ-type (IPR000433) and a C-terminal UBA (IPR015940), all highly conserved when compared to the *D. melanogaster* single isoform of p62 (Ref(2)P) (Fig 2A). The 3D model of the *R. prolixus* p62 protein was predicted using the fold recognition bioinformatics tool I-TASSER (Fig 2B). RT-qPCR showed that expression levels of the *R. prolixus* p62 in vitellogenic females are similar in the fat body, midgut and ovary dissected from vitellogenic females (Fig 2B). Within the

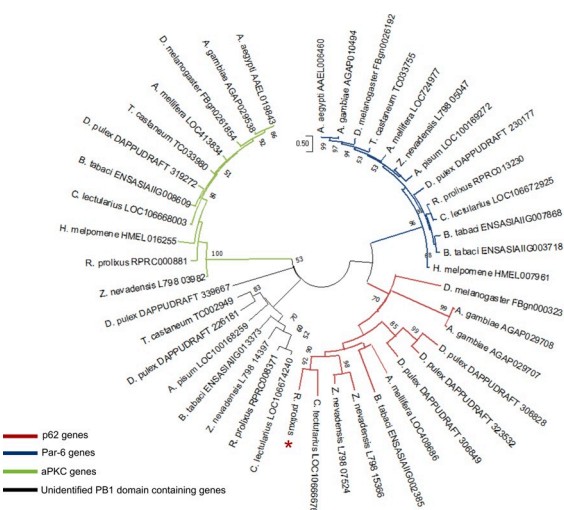

**Fig 1. *R. prolixus* p62 phylogenetic analysis.** Protein sequences with Pfam domain PF00564 (PB1 domain) from different species were aligned using ClustalW, and the dendrogram was constructed by the maximum likelihood method. Bootstrap values are indicated in branches when greater than 50, and the bar indicates substitutions per site. Blue clade: par-6 genes; Red clade: p62 genes (Asterisk: *Rhodnius prolixus* p62 gene). Black clade: unidentified PB1 domain containing protein genes; Green clade: aPKC genes.

ovaries, the levels of p62 were also similar in the tropharium (the structure where the germ cell cluster and the nurse cells are located) and developing oocytes (pre vitellogenic, vitellogenic, choriogenic and chorionated) (Fig 2C). This data indicates that p62 is systemically expressed in somatic and germline tissues of this vector.

## Parental RNAi silencing of p62 is efficient but leads to no apparent changes in digestion, lifespan, oviposition and embryo viability

We synthesized a specific double stranded RNA designed to target the sequence of *R. prolixus* p62. dsRNAs were directly injected into the female's hemocoel two days before the blood meal. RT-qPCR showed that p62 knockdown was efficient, with an average of 90% of mRNA silencing in the ovaries and midgut, and 70% in the fat body of vitellogenic females at days 7, 14 and 21 days after the blood feeding (Fig 2D). Although efficient, the silencing of p62 did not result in major alterations in the insect's blood meal digestion—indirectly measured by the insect's weight after the blood meal (Fig 3A) and protein quantifications at day 7 after the blood meal (Fig 3B). The insects' overall pattern and levels of yolk protein in the hemolymph were also not affected (Figs 3C and 3D and S3), as well as oviposition (Fig 3F and 3G), embryo viability rates (Fig 3H) and parental lifespan (Fig 3E). The eggs laid by control and silenced females were observed under the stereoscope (Fig 4A) and no changes in their external morphology ultrastructure were observed as seen by SEM (Fig 4C). In addition, the overall pattern of size and complexity of the yolk organelles obtained from chorionated oocytes was observed using flow cytometry, as previously described by [23], and no alterations were observed between control (dsMal) and silenced samples (Fig 4B).

Altogether, the absence of apparent phenotypes points to 1) the possibility that although systemically expressed in the adult insect, the function of p62 might be nonessential for the general insect physiology under our in-house conditions; 2) the mRNA silencing, although efficient, did not result in protein down regulation due to high stability and/or low turnover

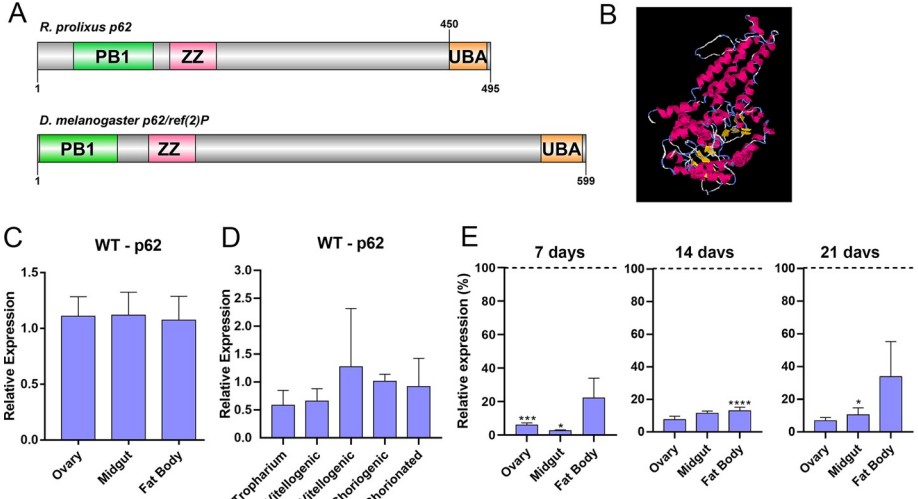

**Fig 2. *R. prolixus* p62 is conserved and systemically expressed in vitellogenic females. A.** Schematic diagram comparing the p62 protein from *R. prolixus* and *D. melanogaster* with their conserved domains. Graphics generated using the DOG 2.0 software and conserved domains predicted using Pfam. PB1 (IPR000270), Znf_ZZ-type (IPR000433) and UBA (IPR015940). **B.** protein three-dimensional model predicted using I-TASSER. **C.** RT-qPCR showing gene expressions in different organs of vitellogenic females. One-Way ANOVA p>0.05. **D.** RT-qPCR showing gene expressions in the different components of the ovary: Tropharium, previtellogenic follicles, vitellogenic follicles, choriogenic follicles and chorionated oocytes. One-Way ANOVA p>0.05. Expression was calculated using the delta $C_t$ method and plotted as $2^{-dCT}$. **E.** The levels of p62 expression in different organs of vitellogenic females were quantified at 7, 14 and 21 days after blood feeding. *p< 0,05, ***p< 0,0001. Student's t-test. The control (dsMal) is represented by the dotted lines, and each sample percentage expression is represented by the bars. All graphs show mean ± SEM (n = 4–7).

rates; and 3) silencing of p62 triggered compensatory mechanisms that allowed the insect to adapt and accomplish general physiology tasks as efficiently as control individuals.

Regarding hypothesis number one, one should consider the possibility that the function of p62 might be necessary under conditions different from those in which we normally keep our insects. Changes in environmental conditions (temperature, humidity, photoperiod, etc.), nutritional status (different cycles and types of diet), immunological stress (exposure to pathogens, symbionts, etc.), among many other factors, can alter the general physiology of individuals at the molecular level, exposing the specific function of a certain gene [58–60]. In this case, we cannot rule out the hypothesis that p62 function can emerge as required under a certain condition that is not met under our in-house conditions.

To address the second hypothesis, accessing the p62 protein levels using reliable antibodies would be vital for improved interpretations of our findings. Still, our experimental options are limited to the methods available for a non-model organism such as *Rhodnius prolixus*, and, in our hands, commercial cross-species antibodies usually do not generate consistent results. For this reason, we ordered custom made antibodies against *R. prolixus* p62 raised against a synthetic N-terminal peptide of 14 AAs of the *R. prolixus* p62 protein. The final bleed, with an ELISA title of 1:512.000, was used for immunoblottings with our samples in ATG6- [22] and ATG8-silenced [50] insects, as well as p62-silenced samples. Because p62 is typically used as read out of autophagic flux, it is expected that the silencing of ATGs would result in an increase in the p62 fragment (indicating decreased autophagic flux) [6]. Unfortunately, the raised antiserum labeled two bands, and none of them matched the expected *R. prolixus* p62 molecular weight of approximately 55 kDa. Furthermore, none of the gene-silencings

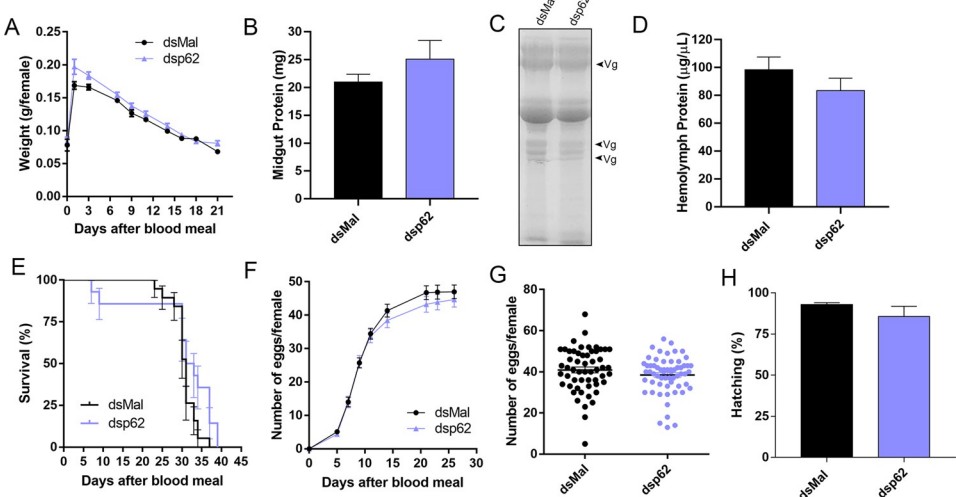

**Fig 3. Silencing of p62 does not affect longevity, digestion, yolk protein levels, oviposition and embryo viability.**
**A.** Control and silenced females were weighted for different days after the blood meal (n = 30). Two-Way ANOVA, p>0.05. **B.** Midgut protein quantifications at day 7 after the blood meal in control and p62 dsRNA-injected insects (n = 3–5). Student's t-test p>0.05. **C.** Hemolymph protein profiles shown in a 10% SDS-PAGE for control and silenced females. Arrowheads point to the main yolk protein vitellogenin (Vg) (n = 4). Representative image. **D.** Hemolymph protein quantifications at day 7 after the blood meal (n = 5). Student's t-test p>0.05. **E.** Survival rates of control and p62 dsRNA-injected insects (n = 30). p>0.05, Log-rank (Mantel-Cox) test. **F.** Oviposition rates of control and silenced females over the gonotrophic cycle (n = 30). Two-Way ANOVA, p>0.05. **G.** Total oviposition of control and silenced females (n = 30). Student's t-test, p>0.05. **H.** Hatching rates of the F1 embryos from control and silenced females (n = 30). Chi-squared test, p>0.05. All graphs show mean ± SEM.

described above resulted in alterations in the detected bands, pointing to unspecific labeling (S4 Fig). Thus, unfortunately, we could not follow through with the antibodies and the direct testing of this hypothesis. Nevertheless, even while we can't rule out the hypothesis that the p62 protein was not downregulated, it is crucial to note that the persistent mRNA silencing (until 21 days after the blood feeding, 23 days after the dsRNA injection, Fig 1D) would require high levels of protein stability to sustain function for over the 21 days of gonotrophic cycle.

Regarding the third hypothesis, of compensatory mechanisms, it is known that UPS and autophagy crosstalk to coordinate intracellular proteostasis [61, 62]. Thus, we decided to test the levels of members of the autophagy and UPS pathways, in the ovaries of control (dsMal) and p62-silenced insects. Results show that the mRNAs levels of ATG1, ATG6 and polyubiquitin (pUBQ) were 7 x, 3 x and 2.5 x increased in silenced ovaries when compared to their control levels (Fig 5A). On the other hand, among the different isoforms of the ovary-expressed UPR chaperones BiPs and PDIs [20, 63], only BiP2 and BiP4 presented slight tendencies in up regulations (Fig 5B), suggesting that some compensatory mechanism among autophagy and UPS were triggered to allow adaptations to the p62-deficiency.

## Knockdown of p62 in insects previously silenced for the E1-ubiquitin conjugating enzyme did not result in cumulative phenotypes

Although we found no apparent phenotypes in p62-silences insects, we decided to further investigate its potential role under an alternative condition, where the presence and/or up-regulation of p62 might play a role in adapting to changes in cellular proteostasis. In a previous study, our team discovered that inhibiting ubiquitin enzymes, particularly the E1-conjugating enzyme, causes the ovary to up-regulate p62, indicating a p62-dependent partnership between

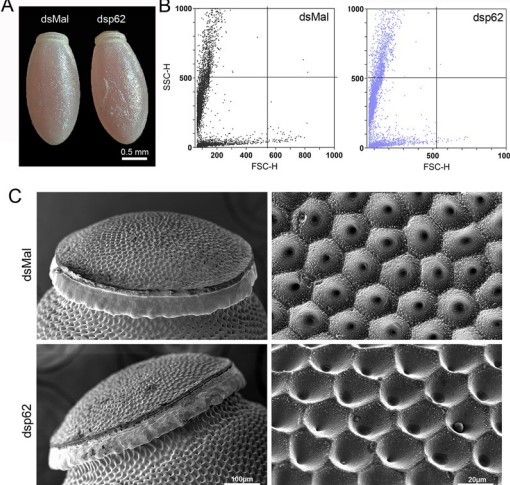

**Fig 4. Eggs generated by p62 silenced females do not present changes in morphology and yolk organelles biogenesis. A.** Representative images of freshly laid (0-96h) control and silenced eggs observed under the stereoscope. **B.** Flow cytometry profiles of the yolk organelles extracted from choriogenic oocytes of control and silenced females. Representative plots of 5 oocytes dissected from different insects (n = 5). **C.** Freshly laid eggs were processed for scanning electron microscopy to access the ultrastructure of their chorion outer surface. Left panel: Detail of the anterior pole of control and silenced eggs showing the structure of the operculum. Right panel: Higher magnification images of the exochorion outer surface of the control and silenced eggs. Images are representative of at least 5 eggs laid by different insects.

UPS and autophagy in maintaining protein homeostasis under those circumstances [24]. Furthermore, in mammalian cells, it has been previously observed that proteasome inhibition caused the induction of p62 and ATG8/GABARAP [64]. Thus, we hypothesized that in E1-silenced insects, where the levels of p62 are endogenously inducted, the effects of silencing p62 would be more prominently observed. To test this, we performed double silencing of the E1-conjugating enzyme and p62 and observed the generated phenotypes. We found that, in the ovary, silencing of p62 did not alter the levels of E1 mRNA transcripts (Fig 6A), whereas the levels of p62 are highly upregulated when the insect is silenced for E1, as previously observed [24] (Fig 6B). The silencing efficiencies in the ovaries of insects injected with both dsRNAs (dsE1 + dsp62) were of approximately 95% for E1 and 50% for p62 (Fig 6C).

Despite the silencing efficiencies, E1- and p62-double silenced insects presented no changes in digestion and yolk proteins in the hemolymph (Fig 7A–7D). As previously observed, E1-silenced females presented lower lifespan, a 90% decrease in oviposition and embryo viability rates [24] (Fig 7F–7H). Interestingly, double-silenced insects presented slightly higher levels of oviposition and embryo lethality when compared to E1-silenced samples (Fig 7F–7H). The morphology of the eggs laid by double-silenced females were observed and only the phenotypes previously observed for E1 were detected (Fig 8A–8F).

In addition, double silenced samples presented induction of ATG1, ATG6 and pUbq in the ovary at similar levels to the ones observed for p62-silenced females (Fig 9A), as well as tendencies of up regulations of the UPR chaperones BiP1, BiP2 and BiP4 (Fig 9B). Altogether, the absence of cumulative phenotypes in double-silenced insects, when compared to the ones observed for E1-silenced individuals, suggests that the upregulation of p62 in E1-silenced samples is also not necessary to allow adaptations to the E1-caused deficiencies. Experiments designed to identify other selective autophagy proteins, such as NBR1, are currently being performed in our laboratory and further studies focusing on the possibility that p62 function is

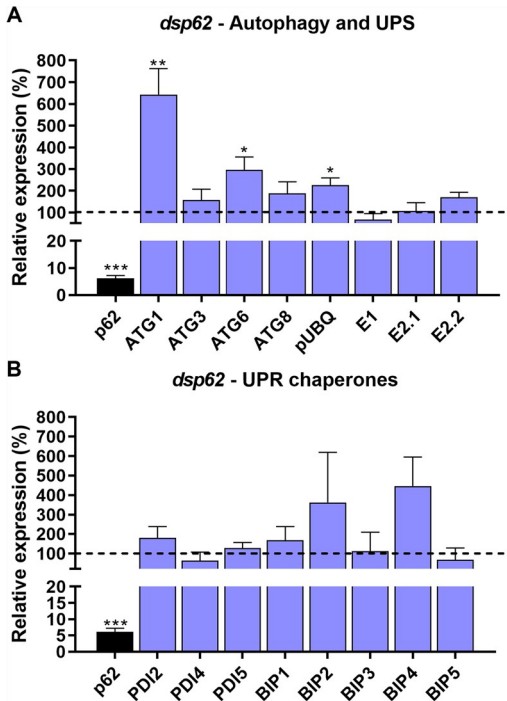

**Fig 5. Silencing of p62 triggers the modulation of autophagy-related genes, genes of the UPS machinery and ER chaperones.** Ovaries dissected from silenced females were tested for the expression levels ATGs, UPS genes, and different isoforms of the UPR chaperones BiPs and PDIs. **A.** Expression levels of different ATGs and UPS genes in the ovaries of females silenced for p62. **B.** Expression levels of different UPR chaperones genes in the ovaries of females silenced for p62. The control (dsMal) is represented by the dotted lines, and each sample percentage expression is represented by the bars. Graphs show mean ± SEM. (n = 4–6) *p< 0.01, **p< 0.005, ***p< 0.001. Student's t-test.

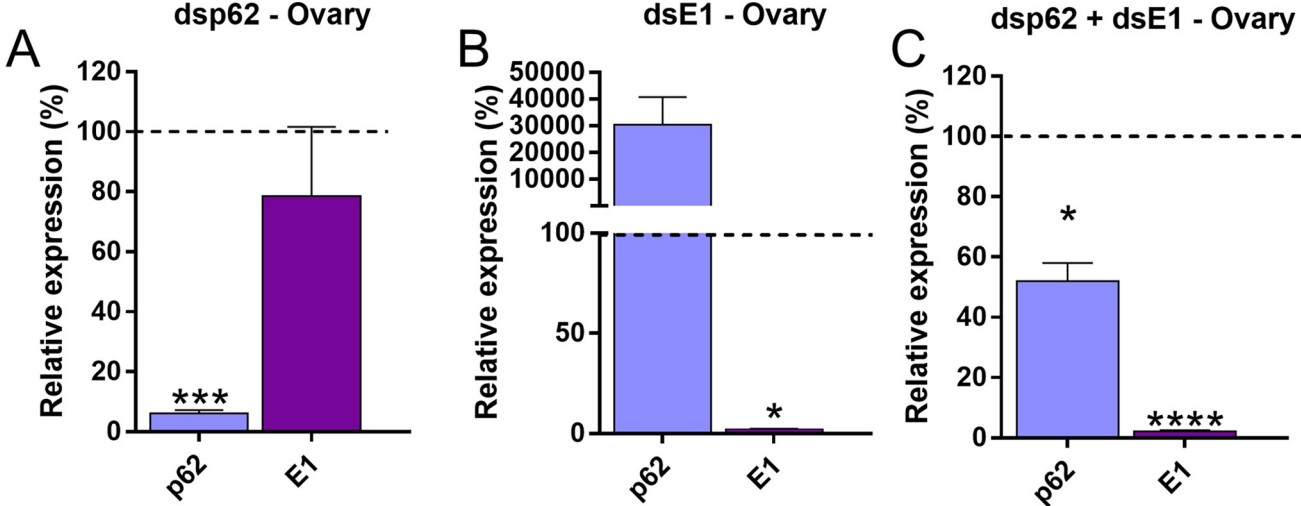

**Fig 6. Silencing efficiencies of individuals injected with dsRNAs targeting p62 and E1.** A. Relative levels of p62 and E1 in the ovaries of females silenced only for p62. **B.** Relative levels of p62 and E1 in the ovaries of females silenced only for E1. **C.** Relative levels of p62 and E1 in the ovaries of females silenced for E1 and p62. The control (dsMal) is represented by the dotted lines, and each sample percentage expression is represented by the bars. Graphs shows mean ± SEM (n = 7). *p< 0.05, ***p< 0.001, ****p< 0.0001, Student's t-test. The results derived from E1-silencing were previously published [65] and the experiments were reproduced in this work.

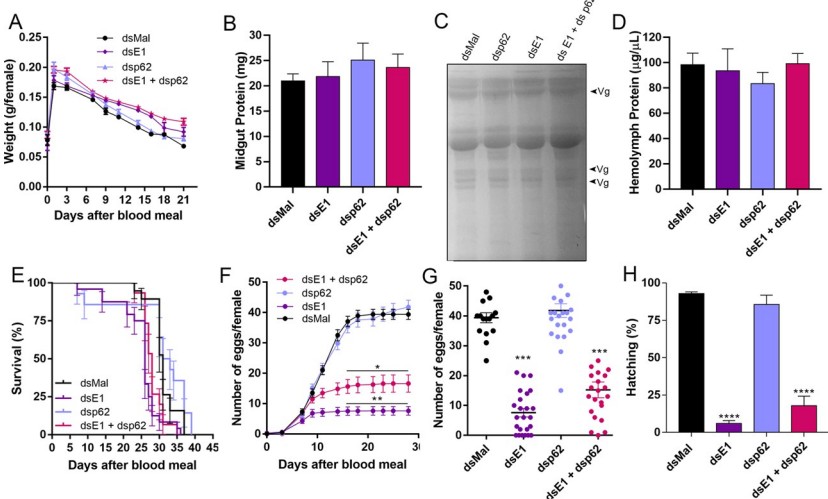

**Fig 7. Double silencing of E1 and p62 does change the physiology phenotypes observed for insects single silenced for E1. A.** Control and silenced females were weighted for different days after the blood meal (n = 30). Two Way ANOVA, p>0.05. **B.** Midgut protein quantifications at day 7 after the blood meal in control and p62 dsRNA-injected insects (n = 3–5). One Way ANOVA, p>0.05. **C.** Hemolymph protein profiles shown in a 10% SDS-PAGE for control and silenced females. Arrowheads point to the main yolk protein vitellogenin (Vg) (n = 4). Representative image. **D.** Hemolymph protein quantifications at day 7 after the blood meal (n = 5). One Way ANOVA, p>0.05. **E.** Survival rates of control and p62 dsRNA-injected insects (n = 30). Log-rank (Mantel-Cox) test), p>0.05. **F.** Oviposition rates of control and silenced females over the gonotrophic cycle (n = 30). Two Way ANOVA, *p< 0.05, **p< 0.01. **G.** Total oviposition of control and silenced females (n = 30). One Way ANOVA, ***p< 0.001. **H.** Hatching rates of the F1 embryos from control and silenced females (n = 30). Chi-squared test, ****p< 0.0001. All graphs show mean ± SEM. The results derived from E1-silencing were previously published [65] and the experiments were reproduced in this work.

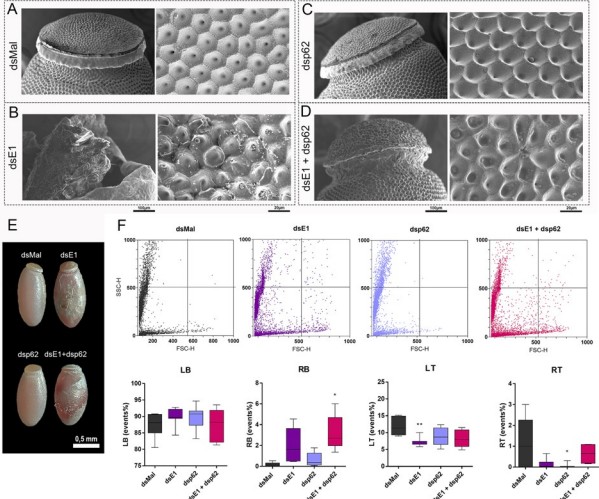

**Fig 8. Eggs generated by E1 and p62 double silenced females do not present changes in chorion ultrastructural morphology and yolk organelles biogenesis. A-D.** Freshly laid eggs were processed for scanning electron microscopy to access the ultrastructure of their chorion outer surface. Right panels: Detail of the anterior pole of control and silenced eggs showing the structure of the operculum. Left panels: Higher magnification images of the exochorion outer surface of the control and silenced eggs. Images are representative of at least 5 eggs laid by different insects (n = 5). **E.** Representative images of freshly laid (0-24h) control and silenced eggs observed under the stereoscope. **F.** Upper panel: Flow cytometry FSC x SSC dot-plots of the yolk organelles extracted from chorionated oocytes obtained from control and silenced females. The plots are representative of five experiments (n = 5). Lower panel: Quantification of the organelles (events) frequency in each quadrant of the plots shown in F (RT, Right top; LT, left top; RB, right bottom; LB, left bottom). *p< 0.05,**p< 0.01, One Way ANOVA. The results derived from E1-silencing were previously published [65] and the experiments were reproduced in this work.

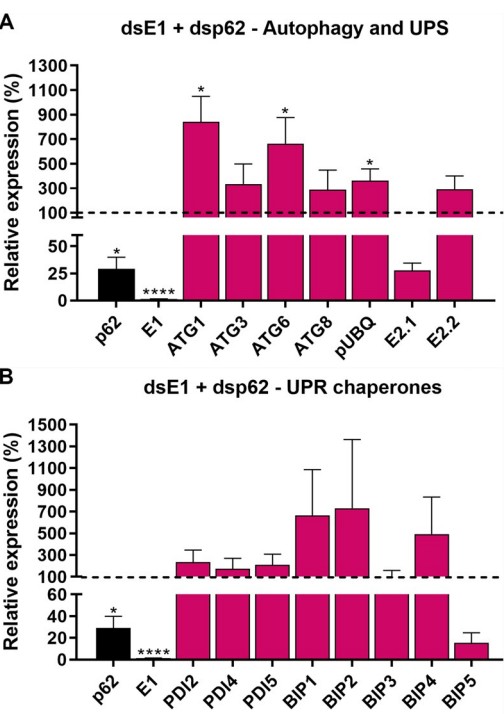

**Fig 9. Double silencing of E1 and p62 triggers the modulation of autophagy-related genes, genes of the UPS machinery and ER chaperones.** Ovaries dissected from silenced females were tested for the expression levels ATGs, UPS genes, and different isoforms of the UPR chaperones BiPs and PDIs. **A.** Expression levels of different ATGs and UPS genes in the ovaries of females silenced for p62. **B.** Expression levels of different UPR chaperones genes in the ovaries of females silenced for p62. The control (dsMal) is represented by the dotted lines, and each sample percentage expression is represented by the bars. Graphs show mean ± SEM. (n = 4–7) *p< 0.01, ****p< 0.0001, Student's t-test.

compensated by a different protein accomplishing a similar function will likely help to unravel the role of the selective autophagy adaptor protein p62 in this model.

## Supporting information

**S1 Table. Genes and primers list.** All sequences were obtained from *Vector Base* (https://www.vectorbase.org/) and primers were synthesized by Macrogen or IDT technologies. (DOCX)

**S1 Fig. Schematic diagram displaying the timeline of dsRNA injections and phenotype observations of our experimental design.** Mated females of the second or third blood feeding after emerging as adults were injected with dsRNAs 2 days before their blood feeding and the generated phenotypes were observed over the following gonotrophic cycle. (TIF)

**S2 Fig. The gene encoding p62 in *R. prolixus*. A.** Schematic diagram showing exons and introns were generated using the exon-intron graphic maker available at wormweb.org/exonintron. Sequence information was obtained from the raw genome data at Vector Base (https://www.vectorbase.org/). **B.** Schematic diagram showing conserved domains and primers targeting regions. The graphic was generated using the DOG 2.0 software (https://dog.biocuckoo.org). Conserved domains were identified using Pfam 35.0 (https://pfam.xfam.org/).

**C.** Predicted model of *R. prolixus* p62 protein using the software ITASSER.
(TIF)

**S3 Fig. Original images of the hemolymph SDS-PAGEs.** Hemolymph protein profiles shown in a 10% SDS-PAGE for control and silenced females. Arrowheads point to the main yolk protein vitellogenin (Vg) (n = 4). MWM, molecular weight markers in kDa. Egg, 30˚g of protein extracted from eggs laid 24h after oviposition. Arrowheads point to the vitellogenin subunits.
(TIF)

**S4 Fig. Original images of the immunoblottings against *R. prolixus* p62.** A 14-amino acid peptide of *R. prolixus* p62 was synthesized and injected in rats. The final bleed was used for immunoblottings against ATG6- and ATG8-silenced samples, as well as p62-silenced samples. Each panel represents a different experiment (N = 4). The numbers on the left indicate the molecular weight marker indications in kDa.
(TIF)

## Acknowledgments

The authors thank CENABIO-UFRJ for providing electron microscopy facilities and Bruna Afonso and Geane Braz for the excellent technical assistance in our insectarium.

## Author Contributions

**Conceptualization:** Isabela Ramos.

**Data curation:** Jéssica Pereira, Samara Santos-Araujo, Larissa Bomfim, David Majerowicz, Attilio Pane.

**Formal analysis:** Jéssica Pereira, Samara Santos-Araujo, Larissa Bomfim, Katia Calp Gondim, David Majerowicz, Attilio Pane, Isabela Ramos.

**Funding acquisition:** Isabela Ramos.

**Supervision:** Isabela Ramos.

**Writing – original draft:** David Majerowicz, Isabela Ramos.

**Writing – review & editing:** Jéssica Pereira, Samara Santos-Araujo, Larissa Bomfim, Katia Calp Gondim, David Majerowicz, Attilio Pane, Isabela Ramos.

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
