## [Decision Letter · Decision Letter 0]

2 Feb 2023

PONE-D-22-35091Functional characterization of p62/SQSTM1 in the vector Rhodnius prolixus reveals a high degree of sequence conservation but no apparent deficiency-related phenotypes in vitellogenic femalesPLOS ONE

Dear Dr. Ramos,

Thank you for submitting your manuscript to PLOS ONE. After careful consideration, we feel that it has merit but does not fully meet PLOS ONE’s publication criteria as it currently stands. Therefore, we invite you to submit a revised version of the manuscript that addresses the points raised during the review process.

We look forward to receiving your revised manuscript.

Kind regards,

Joshua B. Benoit

Academic Editor

PLOS ONE

Journal Requirements:

https://onlinelibrary.wiley.com/doi/10.1002/jcp.30806

In your revision ensure you cite all your sources (including your own works), and quote or rephrase any duplicated text outside the methods section. Further consideration is dependent on these concerns being addressed.

4. Please expand the acronym “JCNE-FAPERJ, INCT-EM-CNPq/FAPERJ, CAPES” (as indicated in your financial disclosure) so that it states the name of your funders in full.

"The authors thank CENABIO-UFRJ and the funding agencies FAPERJ, CNPq and CAPES"

"This research was funded by the following grants: JCNE-FAPERJ (www.faperj.br/), INCT-EM-CNPq/FAPERJ (http://cnpq.br/) and CAPES (www.capes.gov.br/) to I.R.

The funders had no role in study design, data collection and analysis, decision to publish,

or preparation of the manuscript."

7. Please upload a new copy of Figure 1 and 8 as the detail is not clear. Please follow the link for more information:

https://blogs.plos.org/plos/2019/06/looking-good-tips-for-creating-your-plos-figures-graphics/

https://blogs.plos.org/plos/2019/06/looking-good-tips-for-creating-your-plos-figures-graphics/

**Additional Editor Comments:**

The reviewers have indicated that there are considerable issues that must be addressed before reconsideration. This will likely require a combination of extensive editing and some additional analyses. Importantly, the reviewers had a split decision on acceptance (one suggested revision and the other suggested reject), so the manuscript will be sent for review again after revision.

Reviewers' comments:

Reviewer's Responses to Questions

**Comments to the Author**

1. Is the manuscript technically sound, and do the data support the conclusions?

Reviewer #1: Partly

Reviewer #2: Partly

2. Has the statistical analysis been performed appropriately and rigorously? 

Reviewer #1: Yes

Reviewer #2: Yes

3. Have the authors made all data underlying the findings in their manuscript fully available?

Reviewer #1: Yes

Reviewer #2: Yes

4. Is the manuscript presented in an intelligible fashion and written in standard English?

Reviewer #1: Yes

Reviewer #2: Yes

5. Review Comments to the Author

Reviewer #1: General comments:

The manuscript by Pereira et al. studies the autophagy/UPS adaptor protein p62 in the hematophagous model insect Rhodnius prolixus, a vector of Chagas disease. To do so, the authors employed biochemical and molecular biological approaches, including gene silencing. The article addresses an interesting topic, as little information is available on p62 in insects, apart from what has been reported in Drosophila. The manuscript is well organized and written. The objectives and experimental design are clear and straightforward. Broadly, the article is divided into two parts, the first descriptive and the second functional. Although the gene description (sequence, domain, phylogenetic analysis) is well done, I am concerned that the manuscript fails to attribute a functional role to p62 as the evidence obtained is negative. The authors should have performed additional experiments explaining why silencing the p62 gene did not produce any observable phenotype.

Specific comments:

Pages 11-12, section "Parental RNAi silencing of p62...": The authors mention three possible explanations for why the silenced insects showed no change in phenotype, but did not pursue those options. A quick Internet search shows several major antibody suppliers providing anti-p62 polyclonal options that could cross-react against the Rhodnius protein. There is even an antibody that is described as reacting against "invertebrate" p62 and is therefore worth testing (https://www.novusbio.com/products/p62-sqstm1-antibody_nbp1-48320).

Since the title of the article reports that this is a "functional characterization" of the p62, more experiments that attempt to shed light on the function of the protein should be included.

- Page 8, first paragraph: citations are missing. Please include them.

- Page 10, first line: "melpomene" should not be in capital letters. Please correct.

- Page 10, second line: "lectularius" is misspelled. Please correct.

- Page 10, third line: "Daphnia" is misspelled. Please correct.

- Table S1: The word "list" should not be capitalized. Please correct.

- Page 10, RNAi silencing section: There is a space missing between "606" and "bp". Please add. Please revise this aspect (space between magnitudes and their respective units) throughout the manuscript.

- Page 11, "Yolk organelles..." section: No need to capitalize "flow cytometry" in the title.

- Fig. 1 has low resolution. Please upload an improved image.

- Page 11, "The single isoform..." section, last paragraph: "R. prolixus" needs to be italicized. Please correct.

Reviewer #2: The authors propose the genomic and transcriptional characterization of the protein p62 and the study of its relevance during the vitellogenesis of the Chagas ‘disease vector Rhodnius prolixus.

In my opinion, the manuscript only resolves the aspect related with the genomic characterization of the protein. Moreover, the phylogenetic analysis is not representative. The authors claim that the identity of the protein is confirmed based on its occurrence in same clade with Drosophila melanogaster but, at the same time is true that Daphnia pulex (Crustacea) appears in the clade more proximal to the hemipteran species than D. melanogaster and Anopheles gambiae. Furthermore, it is not clear the reasons why the authors include in the phylogenetic analysis other related proteins. It seems to be not necessary. Otherwise, each one of these analysis involving other proteins, if are trully relevant, should be performed independently of the P62 analysis. Finally, the trees in the current form should not be included.

Furthermore, the authors claim that the gene is absent in Aedes aegypti and other species pertaining to other insect orders. It seems not probable the the protein could be lost in Ae. aegypti, being present in the related genus Anopheles. Indeed, it seems not probable that this protein could be lost along insect evolution. It would more probable that the authors didn´t find the corresponding sequences in the genomes.

Regarding the experimental designs, a number of situations are not clear. Why the authors used females after two or three feeding cycles. Why didn´t use females on the first cycle?

I assume that on the first cycle they can get a more accurate standardization of the individuals used for experiments. It is also not clear, when the silencing treatments were done. RNAi silencing treatment of unfed females means first blood feeding as adults as it can be interpreted from M&M section? It means that experimental data were taken after more than 30 days after treatment? Any way, if the females were fed three times, is possible to analyse them in a synchronized physiological condition?

The authors also claim that p62 might be not relevant under in-house conditions! Is this possible? Are the physiology of the insects different in house-conditions?

Minor comments:

- “Heliconius Melpomene” must be “Heliconius melpomene”

- “Cimex letularius” is “Cimex lectularius”

- “Daphia pulex” should be changed by “Daphnia pulex”

Finally, we found that the manuscript present both experimental and theoretical failures to be published in the current form.

6. PLOS authors have the option to publish the peer review history of their article (what does this mean?). If published, this will include your full peer review and any attached files.

Reviewer #1: No

Reviewer #2: No

---

## [Author Response · Author response to Decision Letter 0]

21 Mar 2023

Reviewer #1: General comments:

The manuscript by Pereira et al. studies the autophagy/UPS adaptor protein p62 in the hematophagous model insect Rhodnius prolixus, a vector of Chagas disease. To do so, the authors employed biochemical and molecular biological approaches, including gene silencing. The article addresses an interesting topic, as little information is available on p62 in insects, apart from what has been reported in Drosophila. The manuscript is well organized and written. The objectives and experimental design are clear and straightforward. 

Author´s answer: We thank the reviewer for the careful revision and for the positive appreciation.

Broadly, the article is divided into two parts, the first descriptive and the second functional. Although the gene description (sequence, domain, phylogenetic analysis) is well done, I am concerned that the manuscript fails to attribute a functional role to p62 as the evidence obtained is negative. The authors should have performed additional experiments explaining why silencing the p62 gene did not produce any observable phenotype.

Author´s answer: We have added new experiments and discussion regarding the absence of apparent phenotypes. Please see the information below. 

Specific comments:

Pages 11-12, section "Parental RNAi silencing of p62...": The authors mention three possible explanations for why the silenced insects showed no change in phenotype, but did not pursue those options. A quick Internet search shows several major antibody suppliers providing anti-p62 polyclonal options that could cross-react against the Rhodnius protein. There is even an antibody that is described as reacting against "invertebrate" p62 and is therefore worth testing (https://www.novusbio.com/products/p62-sqstm1-antibody_nbp1-48320).

Author´s answer: Agreed. Regarding the antibodies, we failed to add to the original manuscript the data that we already had with custom-made antibodies raised against R. prolixus p62. Please see below. Regarding the other hypotheses, we did try to pursue them with the available tools for this model. They are now better discussed in the revised manuscript. All is explained below.

Explanation 1) We cannot rule out the possibility that indeed there is no apparent deficiency phenotypes arising from the silencing of this gene under our conditions of insect-rearing and observation. This possibility was better discussed in the revised manuscript. The new text added to the manuscript is copied below:

“Regarding hypothesis number one, one should consider the possibility that the function of p62 might be necessary under conditions different from those in which we normally keep our insects. Changes in environmental conditions (temperature, humidity, photoperiod, etc.), nutritional status (different cycles and types of diet), immunological stress (exposure to pathogens, symbionts, etc.), among many other factors, can alter the general physiology of individuals at the molecular level, exposing the specific function of a certain gene (Schmidt-Nielsen, 1997; Hoffmann, 2012; Zhang et al., 2019). In this case, we cannot rule out the hypothesis that p62 function can emerge as required under a certain condition that is not met under our in-house conditions.”

Explanation 2) mRNA silencing was efficient (which we tested), but protein decrease was not. To address this possibility, we agree that accessing the protein levels using antibodies would be vital for improved interpretations of our findings, and we did test antibodies raised against a synthetic peptide of R. prolixus p62. The data and its methodology was included in the revised manuscript as Fig S4. The text (included in the results and discussion section) is copied below:

“To address the second hypothesis, accessing the p62 protein levels using reliable antibodies would be vital for improved interpretations of our findings. Still, our experimental options are limited to the methods available for a non-model organism such as Rhodnius prolixus, and, in our hands, commercial cross-species antibodies usually do not generate consistent results. For this reason, we ordered custom made antibodies against R. prolixus p62 raised against a synthetic N-terminal peptide of 14 AAs of the R. prolixus p62 protein. The final bleed, with an ELISA title of 1:512.000, was used for immunoblottings with our samples in ATG6- (Vieira et al., 2018b) and ATG8-silenced (Pereira et al., 2020) insects, as well as p62-silenced samples. Because p62 is typically used as read out of autophagic flux, it is expected that the silencing of ATGs would result in an increase in the p62 fragment (indicating decreased autophagic flux) (Klionsky et al., 2021). Unfortunately, the raised antiserum labeled two bands, and none of them matched the expected R. prolixus p62 molecular weight of approximately 50 kDa. Furthermore, none of the gene-silencings described above resulted in alterations in the detected bands, pointing to unspecific labeling. Thus, unfortunately, we could not follow through with the antibodies and direct testing of this hypothesis.”

Fig S4. Immunoblotting against R. prolixus p62. Four independent experiments.

In addition, we have performed new qPCRs to test if the silencing of p62 mRNA was persistent until the end of the analyzed gonotrophic cycle (21 days after the blood meal). We found that the p62 mRNA was still silenced after 14 and 21 days after the blood meal. Thus, the hypothesis of residual protein levels impairing the observation of phenotypes would arise from unusual high levels of stability of the p62 protein, and not from a transient mRNA silencing. This data (qPCR - p62 mRNA silencing at days 14 and 21 after the blood meal) was added to the manuscript as Fig 1D and discussed. 

The revised text including the discussion on the new qPCRs is copied below:

“Nevertheless, even while we can't completely rule out the hypothesis that the p62 protein was not downregulated, it is crucial to note that the persistent mRNA silencing (until 21 days after the blood feeding, 23 days after the dsRNA injection, Fig 1D) would require unusual high levels of protein stability to sustain function for over the 21 days of gonotrophic cycle.”

Explanation 3) Silencing of p62 triggered compensatory mechanisms that allowed the insect to adapt and accomplish general physiology tasks as efficiently as control individuals. 

This hypothesis was pursued and discussed by testing the modulation of autophagy and UPR markers.

Since the title of the article reports that this is a "functional characterization" of the p62, more experiments that attempt to shed light on the function of the protein should be included.

Author´s answer: The supplementary experiments to attempt to shed light on p62´s function were described above. However, we agree that the term functional characterization in the title does not match the findings. So, we changed the title to: “Gene identification and RNAi-silencing of p62/SQSTM1 in the vector Rhodnius prolixus reveals a high degree of sequence conservation but no apparent deficiency-related phenotypes in vitellogenic females”.

- Page 8, first paragraph: citations are missing. Please include them.

Author´s answer: Done. The following references were added.

(Nunes-da-fonseca et al., 2017; World Health Organization, 2019; Eberhard et al., 2020; Lange et al., 2022)

- Page 10, first line: "melpomene" should not be in capital letters. Please correct.

Author´s answer: Done.

- Page 10, second line: "lectularius" is misspelled. Please correct.

Author´s answer: Done.

- Page 10, third line: "Daphnia" is misspelled. Please correct.

Author´s answer: Done.

- Table S1: The word "list" should not be capitalized. Please correct.

Author´s answer: Done.

- Page 10, RNAi silencing section: There is a space missing between "606" and "bp". Please add. Please revise this aspect (space between magnitudes and their respective units) throughout the manuscript.

Author´s answer: Done and corrected throughout the revised manuscript.

- Page 11, "Yolk organelles..." section: No need to capitalize "flow cytometry" in the title.

Author´s answer: Corrected.

- Fig. 1 has low resolution. Please upload an improved image.

Author´s answer: Done. We apologize for the poor resolution image.

- Page 11, "The single isoform..." section, last paragraph: "R. prolixus" needs to be italicized. Please correct.

Author´s answer: Corrected.

Reviewer #2: The authors propose the genomic and transcriptional characterization of the protein p62 and the study of its relevance during the vitellogenesis of the Chagas ‘disease vector Rhodnius prolixus. In my opinion, the manuscript only resolves the aspect related with the genomic characterization of the protein. 

Author´s answer: We thank the reviewer for the comments. All points raised were addressed below. The authors would like to note, however, that this manuscript describes the identification of a new gene, its transcription profile in the different organs and during oogenesis of the vector of an important neglected tropical disease, as well as the data resultant from its RNAi silencing. Although we did not observe apparent phenotypes resultant from the gene silencing to allow conclusions regarding this gene functional role, the manuscript presents original research performed to a high technical standard. Thus, respectfully, we consider the data worth of publishing, and we chose the journal Plos One due to its “mission to publish all valid research, considering negative and null results” Please see https://journals.plos.org/plosone/s/criteria-for-publication.

Thus, we appreciate the reviewer´s time to evaluate again the manuscript considering the explanations and corrections in the revised manuscript. 

Moreover, the phylogenetic analysis is not representative. The authors claim that the identity of the protein is confirmed based on its occurrence in same clade with Drosophila melanogaster but, at the same time is true that Daphnia pulex (Crustacea) appears in the clade more proximal to the hemipteran species than D. melanogaster and Anopheles gambiae. 

Author´s answer: We define each clade based on the branches originating at the tree's base, identified with different colors. The high bootstrap values in the blue, red, and green clades provide statistical support for the hypothesis that the proteins in each of these clades are of different origins. On the other hand, for example, the D. pulex DAPPUDRAFT 339667 protein was external to the clades and could not be identified based on other genomes. Therefore, we used the D. melanogaster genome annotations as a guide, as this genome has the most reliable annotation. Based on this, we identified all red clade proteins as p62 orthologs in their respective organisms. The same goes for the other clades. However, statistical support is low within each clade, which does not allow us to state with certainty the relationships between these proteins. 

Furthermore, it is not clear the reasons why the authors include in the phylogenetic analysis other related proteins. It seems to be not necessary. Otherwise, each one of these analysis involving other proteins, if are trully relevant, should be performed independently of the P62 analysis. Finally, the trees in the current form should not be included.

Author´s answer: We included other proteins in the analysis to perform an unbiased approach. Our approach included all PB1 domain proteins, including p62 and others, present in the analyzed genomes. This approach obliges us to analyze all the proteins, preventing the researcher from introducing bias if he searches with non-automated criteria. For example, we could have searched for Blast using a p62 protein as a query and included the hits obtained in the analysis. However, depending on the chosen inclusion criteria, many other unrelated proteins could have been included (if the inclusion criteria were not strict), or we could not discover the duplications seen in A. gambiae and D. pulex (if the inclusion criteria inclusion was too strict). Our approach avoids these types of problems.

Furthermore, the authors claim that the gene is absent in Aedes aegypti and other species pertaining to other insect orders. It seems not probable the the protein could be lost in Ae. aegypti, being present in the related genus Anopheles. Indeed, it seems not probable that this protein could be lost along insect evolution. It would more probable that the authors didn´t find the corresponding sequences in the genomes.

Author´s answer: It is possible that the genomes of some insects do not have the missing proteins annotated, and, by our search strategy, we would not have found them. However, nothing prevents the gene from being lost in A. aedes, and not in A. gambiae. Gene loss has happened several times during insect diversification (Eirín-López et al., 2012). These hypotheses are mentioned in the revised manuscript:

“On the other hand, the gene is apparently absent in A. aegypti, H. melpomene, T. castaneum, and A. pisum. Whether the gene was lost in the evolutionary process or was just not annotated in the genomes needs to be further investigated (Fig 1, red clade).”

Regarding the experimental designs, a number of situations are not clear. Why the authors used females after two or three feeding cycles. Why didn´t use females on the first cycle?

I assume that on the first cycle they can get a more accurate standardization of the individuals used for experiments.

Author´s answer: All females used in this work were obtained from our insectarium where mated females are fed for the first time (as adult insects) in live-rabbit blood 14 days after the 5th instar nymph to adult ecdysis. During the first blood feeding cycle, the insects are kept back generating the eggs that will hatch as first instar nymphs to maintain the insectarium. After the first blood feeding, all adult insects are fed every 21 days and only fully gorged insects are used for the experiments. Thus, females of the second or third blood feeding were used, and they are highly synchronized regarding blood feeding, digestion, and oviposition. Functional, molecular and physiology studies have been performed using these insects over the past decades. A few examples follow (Masuda and Oliveira, 1985; Oliveira et al., 1989, 2000, 2017, 2018; Moreira et al., 2003; Bouts et al., 2007; Walter-Nuno et al., 2013; Sterkel et al., 2016; Brito et al., 2018; Vieira et al., 2018a, 2021; Pereira et al., 2022, 2020b; Rios et al., 2021; Silva-Oliveira et al., 2021; Entringer et al., 2021)

It is also not clear, when the silencing treatments were done. RNAi silencing treatment of unfed females means first blood feeding as adults as it can be interpreted from M&M section? It means that experimental data were taken after more than 30 days after treatment? Any way, if the females were fed three times, is possible to analyse them in a synchronized physiological condition?

Author´s answer: We apologize for not making it clearer. The insects were injected 2 days before their blood feeding (for the second or third time that they were feeding as adults), and the phenotypes were observed over the following gonotrophic cycle. The only data acquired for more than 30 days after the dsRNA injection (28 days after blood feeding) was the survival data.

To make it clearer, we have added the timeline of our experimental design as Fig S1. 

Also, we have added the data of p62 gene silencing at 14 and 21 days after feeding, and found that its silencing efficiency remained unaltered over that period (please see Fig 2D in the revised manuscript). Thus, all phenotypes were analyzed under conditions where p62 mRNA was silenced.

The authors also claim that p62 might be not relevant under in-house conditions! Is this possible? Are the physiology of the insects different in house-conditions?

Author´s answer: We meant that one should consider the possibility that the function of p62 might be necessary under conditions different from those in which we normally keep our insects. Changes in environmental stress conditions (temperature, humidity, photoperiod, etc.), nutritional stress (different cycles and types of diet, in the case of R. prolixus, blood from different hosts, for example), immunological stress (exposure to pathogens, symbionts, etc.) among many other factors, can alter the physiology of individuals exposing the specific function of a certain gene. We have made this point clearer in the discussion of the revised manuscript. The new text is copied below:

“Regarding hypothesis number one, one should consider the possibility that the function of p62 might be necessary under conditions different from those in which we normally keep our insects. Changes in environmental conditions (temperature, humidity, photoperiod, etc.), nutritional status (different cycles and types of diet), immunological stress (exposure to pathogens, symbionts, etc.), among many other factors, can alter the general physiology of individuals at the molecular level, exposing the specific function of a certain gene (Schmidt-Nielsen, 1997; Hoffmann, 2012; Zhang et al., 2019). In this case, we cannot rule out the hypothesis that p62 function can emerge as required under a certain condition that is not met under our in-house conditions”

comments:

- “Heliconius Melpomene” must be “Heliconius melpomene”

Author´s answer: Corrected.

- “Cimex letularius” is “Cimex lectularius”

Author´s answer: Corrected.

- “Daphia pulex” should be changed by “Daphnia pulex”

Author´s answer: Corrected.

Finally, we found that the manuscript present both experimental and theoretical failures to be published in the current form.

Author´s answer: The alleged experimental and theoretical failures were clarified above. 

References:

Bouts, D. M. D., do Amaral Melo, A. C., Andrade, A. L. H., Silva-Neto, M. A. C., de Oliveira Paiva-Silva, G., Sorgine, M. H. F., et al. (2007). Biochemical properties of the major proteins from Rhodnius prolixus eggshell. Insect Biochem. Mol. Biol. 37, 1207–1221.

Brito, T., Julio, A., Berni, M., de Castro Poncio, L., Bernardes, E. S., Araujo, H., et al. (2018). Transcriptomic and functional analyses of the piRNA pathway in the Chagas disease vector Rhodnius prolixus. PLoS Negl. Trop. Dis. 12, e0006760.

Eberhard, F. E., Cunze, S., Kochmann, J., and Klimpel, S. (2020). Modelling the climatic suitability of Chagas disease vectors on a global scale. Elife 9, e52072. doi:10.7554/eLife.52072.

Eirín-López, J. M., Rebordinos, L., Rooney, A. P., and Rozas, J. (2012). The birth-and-death evolution of multigene families revisited. Repetitive DNA 7, 170–196.

Entringer, P. F., Majerowicz, D., and Gondim, K. C. (2021). The fate of dietary cholesterol in the kissing bug Rhodnius prolixus. Front. Physiol. 12, 654565.

Hoffmann, K. H. (2012). Environmental physiology and biochemistry of insects. Springer Science & Business Media.

Klionsky, D. J., Abdel-Aziz, A. K., Abdelfatah, S., Abdellatif, M., Abdoli, A., Abel, S., et al. (2021). Guidelines for the use and interpretation of assays for monitoring autophagy. Autophagy 17, 1–382.

Lange, A. B., Leyria, J., and Orchard, I. (2022). The hormonal and neural control of egg production in the historically important model insect, Rhodnius prolixus: A review, with new insights in this post-genomic era. Gen. Comp. Endocrinol. 321–322. doi:10.1016/j.ygcen.2022.114030.

Masuda, H., and Oliveira, P. L. (1985). Characterization of vitellin and vitellogenin of Rhodnius prolixus. identification of phosphorilated compounds in the molecule . Insect Bichemistry Mol. Biol. 15, 543–550.

Moreira, M. F., Coelho, H. S. L., Zingali, R. B., Oliveira, P. L., and Masuda, H. (2003). Changes in salivary nitrophorin profile during the life cycle of the blood-sucking bug Rhodnius prolixus. Insect Biochem. Mol. Biol. 33, 23–28.

Nunes-da-fonseca, R., Berni, M., Pane, A., and Araujo, H. M. (2017). Rhodnius prolixus: From classical physiology to modern developmental biology. 1–11. doi:10.1002/dvg.22995.

Oliveira, D. S., Brito, N. F., Franco, T. A., Moreira, M. F., Leal, W. S., and Melo, A. C. A. (2018). Functional characterization of odorant binding protein 27 (RproOBP27) from Rhodnius prolixus antennae. Front. Physiol. 9, 1175.

Oliveira, J. H. M., Talyuli, O. A. C., Goncalves, R. L. S., Paiva-Silva, G. O., Sorgine, M. H. F., Alvarenga, P. H., et al. (2017). Catalase protects Aedes aegypti from oxidative stress and increases midgut infection prevalence of Dengue but not Zika. PLoS Negl. Trop. Dis. 11, 1–13. doi:10.1371/journal.pntd.0005525.

Oliveira, M. F., Silva, J. R., Dansa-Petretski, M., de Souza, W., Braga, C. M. S., Masuda, H., et al. (2000). Haemozoin formation in the midgut of the blood-sucking insect Rhodnius prolixus. Febs Lett. 477, 95–98.

Oliveira, P. L., Petretski, M. D. A., and Masuda, H. (1989). Vitelling processing and degradation during embryogenesis of Rhodnius prolixus. Insect Biochem 19, 489–498.

Pereira, J., Dias, R., and Ramos, I. (2022). Knockdown of E1- and E2-ubiquitin enzymes triggers defective chorion biogenesis and modulation of autophagy-related genes in the follicle cells of the vector Rhodnius prolixus. J. Cell. Physiol. 1, 12.

Pereira, J., Diogo, C., Fonseca, A., Bomfim, L., Cardoso, P., Santos, A., et al. (2020). Silencing of RpATG8 impairs the biogenesis of maternal autophagosomes in vitellogenic oocytes, but does not interrupt follicular atresia in the insect vector Rhodnius prolixus. PLoS Negl. Trop. Dis. 14, e0008012. doi:10.1371/journal.pntd.0008012.

Rios, T., Bomfim, L., and Ramos, I. (2021). The transition from vitellogenesis to choriogenesis triggers the downregulation of the UPR sensors IRE1 and PERK and alterations in the ER architecture in the follicle cells of the vector Rhodnius prolixus. Cell Tissue Res. doi:10.1007/s00441-021-03547-z.

Schmidt-Nielsen, K. (1997). Animal physiology: adaptation and environment. Cambridge university press.

Silva-Oliveira, G., De Paula, I. F., Medina, J. M., Alves-Bezerra, M., and Gondim, K. C. (2021). Insulin receptor deficiency reduces lipid synthesis and reproductive function in the insect Rhodnius prolixus. Biochim. Biophys. Acta (BBA)-Molecular Cell Biol. Lipids 1866, 158851.

Sterkel, M., Perdomo, H. D., Guizzo, M. G., Barletta, A. B. F., Nunes, R. D., Dias, F. A., et al. (2016). Tyrosine detoxification is an essential trait in the life history of blood-feeding arthropods. Curr. Biol. 26, 2188–2193.

Vieira, P. H., Benjamim, C. F., Atella, G. C., and Ramos, I. (2021). VPS38/UVRAG and ATG14, the variant regulatory subunits of the ATG6/Beclin1-PI3K complexes, are crucial for the biogenesis of the yolk organelles and are transcriptionally regulated in the oocytes of the vector Rhodnius prolixus. PLoS Negl. Trop. Dis. 1, 1.

Vieira, P. H., Bomfim, L., Atella, G. C., Masuda, H., and Ramos, I. (2018a). Silencing of RpATG6 impaired the yolk accumulation and the biogenesis of the yolk organelles in the insect vector R. prolixus. PLoS Negl. Trop. Dis. 12, e0006507. doi:10.1371/journal.pntd.0006507.

Vieira, P. H., Bomfim, L., Atella, G. C., Masuda, H., and Ramos, I. (2018b). Silencing of RpATG6 impaired the yolk accumulation and the biogenesis of the yolk organelles in the insect vector R. prolixus. PLoS Negl. Trop. Dis. 12. doi:10.1371/journal.pntd.0006507.

Walter-Nuno, A. B., Oliveira, M. P., Oliveira, M. F., Gonçalves, R. L., Ramos, I. B., Koerich, L. B., et al. (2013). Silencing of maternal heme-binding protein causes embryonic mitochondrial dysfunction and impairs embryogenesis in the blood sucking insect Rhodnius prolixus. J. Biol. Chem. 288, 29323–29332.

World Health Organization (2019). WHO. Available at: https://www.who.int/en/news-room/fact-sheets/detail/chagas-disease-(american-trypanosomiasis).

Zhang, D.-W., Xiao, Z.-J., Zeng, B.-P., Li, K., and Tang, Y.-L. (2019). Insect behavior and physiological adaptation mechanisms under starvation stress. Front. Physiol. 10, 163.

---

## [Decision Letter · Decision Letter 1]

18 May 2023

PONE-D-22-35091R1Gene identification and RNAi-silencing of p62/SQSTM1 in the vector Rhodnius prolixus reveals a high degree of sequence conservation but no apparent deficiency-related phenotypes in vitellogenic femalesPLOS ONE

Dear Dr. Ramos,

Thank you for submitting your manuscript to PLOS ONE. After careful consideration, we feel that it has merit but does not fully meet PLOS ONE’s publication criteria as it currently stands. Therefore, we invite you to submit a revised version of the manuscript that addresses the points raised during the review process.

We look forward to receiving your revised manuscript.

Kind regards,

Joshua B. Benoit

Academic Editor

PLOS ONE

Journal Requirements:

Additional Editor Comments (if provided):

The reviewers have suggested a few additional minor comments. After these are addressed, I'll accept the paper.

Reviewers' comments:

Reviewer's Responses to Questions

**Comments to the Author**

1. If the authors have adequately addressed your comments raised in a previous round of review and you feel that this manuscript is now acceptable for publication, you may indicate that here to bypass the “Comments to the Author” section, enter your conflict of interest statement in the “Confidential to Editor” section, and submit your "Accept" recommendation.

Reviewer #1: All comments have been addressed

Reviewer #3: (No Response)

2. Is the manuscript technically sound, and do the data support the conclusions?

Reviewer #1: Yes

Reviewer #3: Partly

3. Has the statistical analysis been performed appropriately and rigorously? 

Reviewer #1: Yes

Reviewer #3: Yes

4. Have the authors made all data underlying the findings in their manuscript fully available?

Reviewer #1: Yes

Reviewer #3: Yes

5. Is the manuscript presented in an intelligible fashion and written in standard English?

Reviewer #1: Yes

Reviewer #3: Yes

6. Review Comments to the Author

Reviewer #1: In the revised version of the MS, the authors have added experiments to improve the articulation between the objective and the hypothesis. They have changed the title to fit the physiological context of the results. In addition, the authors have improved the resolution of the figures. The experimental designs are adequate. Other aspects related to methodology and results have also been satisfactorily addressed. The sample size is adequate to address the hypothesis. Statistical analyses have been correctly applied.

The authors have provided reasonable explanations in the discussion to reinforce the conclusions based on the results obtained. In accordance with suggestions, the discussion section has been greatly improved. Overall, the revised version of the MS has been substantially improved.

Reviewer #3: Pereira and colleagues' work describes the function of the gene that encodes for p62, a receptor involved in selective autophagy of ubiquitinated proteins. Although it is one of the most studied factors involved in autophagy, information on insects is scarce, making the authors' work even more interesting, especially since they use R. prolixus as a model. This triatomine species is one of the most important vectors of Chagas disease, a neglected pathology according to WHO that affects millions of people around the world. The authors first conduct a descriptive analysis of the p62 gene using bioinformatic and molecular analyses, and then a functional analysis using mostly molecular analyses. The manuscript is very well written, with some minor errors that I detail below. My main concern, in line with reviewer 1, is the lack of protein-level analysis of p62. While it would be very interesting to evaluate this protein in this model, the work could be published with a more molecular focus. I detail some suggestions below.

I agree with reviewer one that the authors should better explain why silencing the p62 gene does not produce an observable phenotype. One of these options, as I mentioned before, is to evaluate protein levels either in tissues or at the peripheral level. The authors mention three possible explanations for why phenotypes are not observed: 1) the protein is not essential for the analyzed phenotypes, 2) although silencing was effective, protein levels are not reduced due to various factors, and 3) silencing p62 activates compensatory mechanisms.

Regarding the explanation provided by the authors for hypothesis #1, that potential changes in phenotypes cannot be ruled out if the gene decreases its expression for some reason, this is only true if double-stranded RNA specific to p62 is specifically silenced in nature, which is unlikely.

Regarding the authors' explanation of hypothesis number 2, the authors responded to reviewer 1 that they tested mRNA levels and found that they decreased, but not protein levels. This is inconsistent with the explanation they provide below, as they were not able to measure protein levels, so it is unclear whether the lack of phenotype may be caused by a compensatory mechanism. To test the protein, the authors used a non-commercial antibody, as in their experience, commercial antibodies were not effective. However, their antibody also did not appear to be effective, as the bands observed in Figure S4 did not correspond to the expected molecular weight. Therefore, I suggest not adding this information. To investigate phenotypic changes caused by the lack of a protein, it is necessary to determine whether the levels of that protein have actually been decreased. It is also possible that the correct phenotype or "read out" is not being evaluated. To explain this hypothesis, the authors evaluate through qPCR that gene silencing is maintained over time, which would explain that the lack of observable phenotype is not due to transient silencing. However, it does not fully explain the lack of observable phenotype. They also suggest that this may be due to an unusual stability of the p62 protein. I suggest not using the word "unusual", as protein half-lives can vary widely and conclusions about it cannot be drawn if the protein was not evaluated. I suggest giving the work a more descriptive focus only on the involved gene or evaluating other aspects such as autophagosome formation, as I suggest below. With so little information on p62 in triatomines, I find the information presented in this work by the authors to be very relevant and it would be the starting point for evaluating this protein in more functional analyses in future research. Overall, I think mentioning these hypotheses leaves the reader with the idea that more could have been done, so I suggest not mentioning them if they cannot be explored, as they are very generic and explain what would happen for any gene that is silenced and does not produce observable phenotypes. I believe that the information provided by the authors is sufficient to be published as a descriptive analysis of the p62 gene, without pursuing other objectives that cannot be developed.

Regarding the authors' response to reviewer 1, "We have added new experiments and discussion regarding the absence of apparent phenotypes." The added experiments such as SDS-PAGE and immunoblotting do not fully explain the absence of apparent phenotypes. SDS-PAGE does not clarify whether residual protein levels are sufficient to prevent functional changes. Additionally, the observed bands in immunoblotting do not correspond to the expected weight, thus, no conclusions can be drawn from this experiment. The authors may consider evaluating p62 protein levels using ELISA or at least semi-quantitatively using a functional antibody against p62. As previously suggested, if a functional antibody is not available, the authors may need to change the focus of the manuscript and not pursue the proposed hypotheses. Reviewer 1 also suggested that the authors use a commercial antibody; otherwise, we cannot determine which proposed hypothesis is the most suitable.

Regarding the phrase "Because the UPS and autophagy are essential intracellular degradation routes and p62 is a key mediator that allows direct crosstalk between these pathways, in this work we investigate the silencing effects of p62 in the context of R. prolixus oogenesis." There are kits available for monitoring autophagy, in which p62 is fused to some fluorescent protein. Perhaps the use of these kits could provide more answers about the function of p62 in the relationship between UPS and autophagy. Maybe the authors have already attempted or are attempting to use some of these methods. A quick internet search yielded some results, such as this: https://www.thermofisher.com/order/catalog/product/P36240). Surely the authors can better address this concern than I can.

Later on, the authors present the first functional results. As a suggestion, they could add a three-dimensional modeling to showcase p62, using software for this purpose, such as the I-TASSER package (Yang, J. & Zhang, Y. Protein structure and function prediction using I-TASSER. Curr. Protoc. Bioinform. 52, 5–8 (2015)). It seems to me that this would help the reader to have an idea about the three-dimensional structure of the protein and improve the work aesthetically, but it is not essential for understanding the idea proposed by the authors.

Material and methods

Insects: For the inexperienced reader, could you clarify how they control whether the females have copulated?

References: Are the authors missing in reference 30?

Extraction of RNA and cDNA synthesis: Please clarify which method was used for the elimination of genomic DNA and evaluation of RNA integrity. Typically, DNase and 2% agarose gel are used for RNA integrity.

PCR / RT-qPCR: Did the authors try to check the PCR and qPCR products on an agarose gel? Are there many non-specific bands in the PCR product compared to the qPCR product? I have this question because they use a much higher annealing temperature, 60°C, in qPCR. In this same section, they use the 18S gene as a reference gene. Please use the citation of Majerowicz et al. 2011 and in the use of the 2ΔΔCt method, Livak and Schmittgen, 2011.

RNAi silencing: I recommend that the authors mention the review by Paim et al. 2013. (DOI 10.1111/j.1744-7917.2012.01540.x), which discusses many aspects related to gene silencing in triatomines.

Results and discussion

Section: The single isoform of R. prolixus p62 is conserved within insects and is expressed in the ovaries, developing oocytes, midgut and fat body of adult vitellogenic females. In the 2nd paragraph plese correct: “Reverse transcription quantitative PCR (RT-qPCR) showed that expression levels of the R. prolixus p62 in vitellogenic females are similar in the fat body, midgut and ovary dissected from vitellogenic females.” By “RT-qPCR showed that expression levels of the R. prolixus p62 are similar in the fat body, midgut and ovary dissected from vitellogenic females”.

Section: Parental RNAi silencing of p62 is efficient but leads to no apparent changes in digestion, lifespan, oviposition and embryo viability. It is not very clear what the dotted line corresponding to 100% means. For a better understanding of the reader, I think it would be better to clarify what 100% means and show the expression levels of control individuals, i.e. without the silenced gene. The authors surely have this data. In Figure 3A, the relevance of the experiment is not very clear. I suggest moving it to supplementary data.

Fig 6: If the aim is to show the effect of p62 silencing on E1 and vice versa, please show the levels of p62 and E1 in control animals, i.e., those treated with non-related dsRNA. Again, it would be clearer for the reader if the dashed line representing 100% is clarified and the best way to do so is by showing the expression levels of the control.

What is the difference between Figure 6B and the authors' previous publication? (Pereira J, Dias R, Ramos I. Knockdown of E1- and E2-ubiquitin enzymes triggers defective chorion biogenesis and modulation of autophagy-related genes in the follicle cells of the vector Rhodnius prolixus. J Cell Physiol. 2022;1:12.).

Fig 7. I suggest clarifying in the figure, not only in the text, that the results of the E1 silence were previously published by the authors or only citing them and not showing them in the figure, especially if this treatment does not produce changes in control individuals.

Fig 8. The same suggestion as for Figure 7 applies. The experiments related to E1 were previously published by the authors in Pereira J, Dias R, Ramos I. Knockdown of E1- and E2-ubiquitin enzymes triggers defective chorion biogenesis and modulation of autophagy-related genes in the follicle cells of the vector Rhodnius prolixus. J Cell Physiol. 2022;1:12.

7. PLOS authors have the option to publish the peer review history of their article (what does this mean?). If published, this will include your full peer review and any attached files.

Reviewer #1: No

Reviewer #3: **Yes: **Fabian Orlando Ramos PhD.

Facultad de Ciencias Químicas

Universidad Nacional de Córdoba

---

## [Author Response · Author response to Decision Letter 1]

23 May 2023

R2- Manuscript #PONE-D-22-35091

Reviewer #1: 

In the revised version of the MS, the authors have added experiments to improve the articulation between the objective and the hypothesis. They have changed the title to fit the physiological context of the results. In addition, the authors have improved the resolution of the figures. The experimental designs are adequate. Other aspects related to methodology and results have also been satisfactorily addressed. The sample size is adequate to address the hypothesis. Statistical analyses have been correctly applied. The authors have provided reasonable explanations in the discussion to reinforce the conclusions based on the results obtained. In accordance with suggestions, the discussion section has been greatly improved. Overall, the revised version of the MS has been substantially improved.

Author´s answer: We thank the reviewer for taking the time to read our revised manuscript and for the positive appreciation.

Reviewer #3: 

Pereira and colleagues' work describes the function of the gene that encodes for p62, a receptor involved in selective autophagy of ubiquitinated proteins. Although it is one of the most studied factors involved in autophagy, information on insects is scarce, making the authors' work even more interesting, especially since they use R. prolixus as a model. This triatomine species is one of the most important vectors of Chagas disease, a neglected pathology according to WHO that affects millions of people around the world. The authors first conduct a descriptive analysis of the p62 gene using bioinformatic and molecular analyses, and then a functional analysis using mostly molecular analyses. The manuscript is very well written, with some minor errors that I detail below. My main concern, in line with reviewer 1, is the lack of protein-level analysis of p62. While it would be very interesting to evaluate this protein in this model, the work could be published with a more molecular focus. I detail some suggestions below.

Author´s answer: We thank the reviewer for the positive recommendation.

I agree with reviewer one that the authors should better explain why silencing the p62 gene does not produce an observable phenotype. One of these options, as I mentioned before, is to evaluate protein levels either in tissues or at the peripheral level. The authors mention three possible explanations for why phenotypes are not observed: 1) the protein is not essential for the analyzed phenotypes, 2) although silencing was effective, protein levels are not reduced due to various factors, and 3) silencing p62 activates compensatory mechanisms.

Regarding the explanation provided by the authors for hypothesis #1, that potential changes in phenotypes cannot be ruled out if the gene decreases its expression for some reason, this is only true if double-stranded RNA specific to p62 is specifically silenced in nature, which is unlikely.

Author´s answer: We meant that it is possible that p62 protein is not essential for the phenotypes that we are observing under the described conditions. We could not test protein levels due to the lack of antibodies, so we don´t know if it is decreased or not after RNAi treatment. It is a reasonable possibility that the protein levels did decrease, but this reduction did not generate observable phenotypes, and it has nothing to do with the double-stranded RNA specific to p62 being specifically silenced in nature.

Regarding the authors' explanation of hypothesis number 2, the authors responded to reviewer 1 that they tested mRNA levels and found that they decreased, but not protein levels. This is inconsistent with the explanation they provide below, as they were not able to measure protein levels, so it is unclear whether the lack of phenotype may be caused by a compensatory mechanism. 

Author´s answer: We do not know if the proteins levels were decreased because we did not have the tools to test it. Furthermore, hypotheses #1 and #2 are unrelated possibilities, one OR the other might be happening. 

To test the protein, the authors used a non-commercial antibody, as in their experience, commercial antibodies were not effective. However, their antibody also did not appear to be effective, as the bands observed in Figure S4 did not correspond to the expected molecular weight. Therefore, I suggest not adding this information. 

Author´s answer: The information was included to the revised manuscript to show that, despite their failure, the authors tried to develop methods to study protein-level silencing. We believe it is worthwhile to keep the data available in the publication because the amount of supplemental data is not constrained by the journal, and we appreciate data transparency.

To investigate phenotypic changes caused by the lack of a protein, it is necessary to determine whether the levels of that protein have actually been decreased. It is also possible that the correct phenotype or "read out" is not being evaluated. To explain this hypothesis, the authors evaluate through qPCR that gene silencing is maintained over time, which would explain that the lack of observable phenotype is not due to transient silencing. However, it does not fully explain the lack of observable phenotype. 

Author´s answer: The persistent mRNA silencing (observed by qPCR) does not explain the lack of phenotypes, it only suggests that the absence of phenotypes is not due to a rapid transient silencing that would not result in robust protein decrease. This possibility was discussed.

They also suggest that this may be due to an unusual stability of the p62 protein. I suggest not using the word "unusual", as protein half-lives can vary widely and conclusions about it cannot be drawn if the protein was not evaluated. 

Author´s answer: Agreed. The word unusual was deleted from the sentence.

I suggest giving the work a more descriptive focus only on the involved gene or evaluating other aspects such as autophagosome formation, as I suggest below. With so little information on p62 in triatomines, I find the information presented in this work by the authors to be very relevant and it would be the starting point for evaluating this protein in more functional analyses in future research. Overall, I think mentioning these hypotheses leaves the reader with the idea that more could have been done, so I suggest not mentioning them if they cannot be explored, as they are very generic and explain what would happen for any gene that is silenced and does not produce observable phenotypes. I believe that the information provided by the authors is sufficient to be published as a descriptive analysis of the p62 gene, without pursuing other objectives that cannot be developed.

Author´s answer: We consider mentioning and discussing our thoughts on the hypotheses for not observing apparent phenotypes an important aspect in publishing science.

Regarding the authors' response to reviewer 1, "We have added new experiments and discussion regarding the absence of apparent phenotypes." The added experiments such as SDS-PAGE and immunoblotting do not fully explain the absence of apparent phenotypes. SDS-PAGE does not clarify whether residual protein levels are sufficient to prevent functional changes. Additionally, the observed bands in immunoblotting do not correspond to the expected weight, thus, no conclusions can be drawn from this experiment. The authors may consider evaluating p62 protein levels using ELISA or at least semi-quantitatively using a functional antibody against p62. As previously suggested, if a functional antibody is not available, the authors may need to change the focus of the manuscript and not pursue the proposed hypotheses. Reviewer 1 also suggested that the authors use a commercial antibody; otherwise, we cannot determine which proposed hypothesis is the most suitable.

Author´s answer: The immunoblotting was included, as indicated in the results and discussion, to show our attempt to further investigate p62 protein levels and that, regrettably, we were unable to produce a trustworthy instrument to do so. This data is what we would often refer to as supplementary information.

The original images of the SDS-PAGEs were included as a journal requirement to show original uncropped SDS-PAGEs as supplementary information. They are the original gels that generated Figs 3C and 7C. It is a journal condition (highly supported by the authors) to stimulate data transparency.

Regarding the phrase "Because the UPS and autophagy are essential intracellular degradation routes and p62 is a key mediator that allows direct crosstalk between these pathways, in this work we investigate the silencing effects of p62 in the context of R. prolixus oogenesis." There are kits available for monitoring autophagy, in which p62 is fused to some fluorescent protein. Perhaps the use of these kits could provide more answers about the function of p62 in the relationship between UPS and autophagy. Maybe the authors have already attempted or are attempting to use some of these methods. A quick internet search yielded some results, such as this: https://www.thermofisher.com/order/catalog/product/P36240). Surely the authors can better address this concern than I can.

Author´s answer: It would be interesting to monitor autophagy under the effects of silencing p62. However, such kits use recombinant tagged p62 or LC3 to be transfected into cells and are not suitable for use in live insects. 

Later on, the authors present the first functional results. As a suggestion, they could add a three-dimensional modeling to showcase p62, using software for this purpose, such as the I-TASSER package (Yang, J. & Zhang, Y. Protein structure and function prediction using I-TASSER. Curr. Protoc. Bioinform. 52, 5–8 (2015)). It seems to me that this would help the reader to have an idea about the three-dimensional structure of the protein and improve the work aesthetically, but it is not essential for understanding the idea proposed by the authors.

Author´s answer: Agreed. The p62 model was produced using the suggested software I-TASSER and was included as Fig 2B.

Material and methods

Insects: For the inexperienced reader, could you clarify how they control whether the females have copulated?

Author´s answer: We know because we monitor their oviposition and F1 eclosion rates. Virgin females do not lay many eggs, and the ones that are laid are not viable because they are not fertilized.

This info was added to the methodology. The new text now reads:

“Insects were maintained at a 28 ± 2°C controlled temperature and relative humidity of 70-80%. 

The insects are fed for the first time as adults in live-rabbit blood 14 to 21 days after the 5th instar nymph to adult ecdysis. After the first blood feeding, the insects (male and females) are kept back together to mate and generate the eggs that will hatch as first instar nymphs to maintain the insectarium. We know that the females were mated because we do monitor their oviposition rates and F1 eclosion rates during this cycle (virgin females do not lay many eggs, and they do not hatch because they are not fertilized). After this first blood feeding, all adult females are fed every 21 days and only fully gorged insects (allowed to feed at free demand, usually gaining 6-7 times the insect’s initial body weight in 20-30 min) are used for the experiments. Thus, females of the second or third blood feeding were used, and they are mated, and highly synchronized regarding blood feeding, digestion, and oviposition. All animal care and experimental protocols were approved by the guidelines described in the ethics statement.”

References: 

Are the authors missing in reference 30?

Author´s answer: No. The reference is under the authorship of the “Honeybee Genome Sequencing Consortium”. The reference format was downloaded from the journal link.

Extraction of RNA and cDNA synthesis: Please clarify which method was used for the elimination of genomic DNA and evaluation of RNA integrity. Typically, DNase and 2% agarose gel are used for RNA integrity.

Author´s answer: DNAse treatments and checking RNA integrity in the gels were routinely performed. The info was added to the methodology. The revised text is copied below:

“Reverse transcription reaction was carried out using the High-Capacity cDNA Reverse Transcription Kit (Applied Biosystems) using 1°g of total RNA (after RNase-free DNase I (Invitrogen) treatment and checking RNA integrity in 2% agarose gels), Multiscribe Reverse Transcriptase enzyme (2.5 U/°L) and random primers for 10 min at 25°C followed by 2 hours of incubation at 37°C.”

PCR / RT-qPCR: Did the authors try to check the PCR and qPCR products on an agarose gel? Are there many non-specific bands in the PCR product compared to the qPCR product? I have this question because they use a much higher annealing temperature, 60°C, in qPCR. 

Author´s answer: The PCR products are observed under agarose gels as a first test to access primers functionality and showed no unspecific fragments. In qPCRs we do monitor the reaction melting curves as a read out to the amplification of single specific fragments.

In this same section, they use the 18S gene as a reference gene. Please use the citation of Majerowicz et al. 2011 and in the use of the 2ΔΔCt method, Livak and Schmittgen, 2011.

RNAi silencing: I recommend that the authors mention the review by Paim et al. 2013. (DOI 10.1111/j.1744-7917.2012.01540.x), which discusses many aspects related to gene silencing in triatomines.

Author´s answer: Done. The references were added. 

Results and discussion Section: The single isoform of R. prolixus p62 is conserved within insects and is expressed in the ovaries, developing oocytes, midgut and fat body of adult vitellogenic females. In the 2nd paragraph plese correct: “Reverse transcription quantitative PCR (RT-qPCR) showed that expression levels of the R. prolixus p62 in vitellogenic females are similar in the fat body, midgut and ovary dissected from vitellogenic females.” By “RT-qPCR showed that expression levels of the R. prolixus p62 are similar in the fat body, midgut and ovary dissected from vitellogenic females”.

Author´s answer: Done.

Section: Parental RNAi silencing of p62 is efficient but leads to no apparent changes in digestion, lifespan, oviposition and embryo viability. It is not very clear what the dotted line corresponding to 100% means. For a better understanding of the reader, I think it would be better to clarify what 100% means and show the expression levels of control individuals, i.e. without the silenced gene. The authors surely have this data. 

Author´s answer: To make it clearer, we have added the following sentence to the figure captions:

“The control (dsMal) is represented by the dotted line, and each sample percentage expression is represented by the bars.”

In Figure 3A, the relevance of the experiment is not very clear. I suggest moving it to supplementary data.

Author´s answer: We chose to include Figure 3A because it is an indirect way of showing that the insects display similar rates of blood meal digestion, which is a trigger of the oogenesis program. It is an important information if one aims to study oocyte development and female reproduction in general. 

Fig 6: If the aim is to show the effect of p62 silencing on E1 and vice versa, please show the levels of p62 and E1 in control animals, i.e., those treated with non-related dsRNA. Again, it would be clearer for the reader if the dashed line representing 100% is clarified and the best way to do so is by showing the expression levels of the control.

Author´s answer: Agreed. The sentence below was added to the figure captions:

“The control (dsMal) is represented by the dotted line, and each sample percentage expression is represented by the bars.”

What is the difference between Figure 6B and the authors' previous publication? (Pereira J, Dias R, Ramos I. Knockdown of E1- and E2-ubiquitin enzymes triggers defective chorion biogenesis and modulation of autophagy-related genes in the follicle cells of the vector Rhodnius prolixus. J Cell Physiol. 2022;1:12.).

Author´s answer: No difference. To compare the E1-silencing results with the outcomes of the double silencing treatments (E1 + p62), identical experiments were carried out again in this study (we replicated the E1 data from our prior research). It is crucial to display the E1-silencing data in this work so that the reader may easily compare the outcomes of the three conditions (E1, p62, and E1+p62).

Fig 7. I suggest clarifying in the figure, not only in the text, that the results of the E1 silence were previously published by the authors or only citing them and not showing them in the figure, especially if this treatment does not produce changes in control individuals.

Fig 8. The same suggestion as for Figure 7 applies. The experiments related to E1 were previously published by the authors in Pereira J, Dias R, Ramos I. Knockdown of E1- and E2-ubiquitin enzymes triggers defective chorion biogenesis and modulation of autophagy-related genes in the follicle cells of the vector Rhodnius prolixus. J Cell Physiol. 2022;1:12.

Author´s answer: Agreed. We informed in the figure captions (Figs 6, 7 and 8) that the E1-silencing results were published before. The added text is copied below: 

“The results derived from E1-silencing were previously published (Pereira et al., 2022) and the experiments were reproduced in this work.”

---

## [Editor Report · Decision Letter 2]

6 Jun 2023

Gene identification and RNAi-silencing of p62/SQSTM1 in the vector Rhodnius prolixus reveals a high degree of sequence conservation but no apparent deficiency-related phenotypes in vitellogenic females

PONE-D-22-35091R2

Dear Dr. Ramos,

We’re pleased to inform you that your manuscript has been judged scientifically suitable for publication and will be formally accepted for publication once it meets all outstanding technical requirements.

Kind regards,

Joshua B. Benoit

Academic Editor

PLOS ONE
---

## [Editor Report · Acceptance letter]

14 Jul 2023

PONE-D-22-35091R2 

Gene identification and RNAi-silencing of p62/SQSTM1 in the vector *Rhodnius prolixus* reveals a high degree of sequence conservation but no apparent deficiency-related phenotypes in vitellogenic females 

Dear Dr. Ramos:

I'm pleased to inform you that your manuscript has been deemed suitable for publication in PLOS ONE. Congratulations! Your manuscript is now with our production department. 

Kind regards, 

on behalf of

Dr. Joshua B. Benoit 

Academic Editor

PLOS ONE